# The HER2-directed antibody-drug conjugate DHES0815A in advanced and/or metastatic breast cancer: preclinical characterization and phase 1 trial results

Approved antibody-drug conjugates (ADCs) for HER2-positive breast cancer include trastuzumab emtansine and trastuzumab deruxtecan. To develop a differentiated HER2 ADC, we chose an antibody that does not compete with trastuzumab or pertuzumab for binding, conjugated to a reduced potency PBD (pyrrolobenzodiazepine) dimer payload. PBDs are potent cytotoxic agents that alkylate and cross-link DNA. In our study, the PBD dimer is modified to alkylate, but not cross-link DNA. This HER2 ADC, DHES0815A, demonstrates in vivo efficacy in models of HER2-positive and HER2-low cancers and is well-tolerated in cynomolgus monkey safety studies. Mechanisms of action include induction of DNA damage and apoptosis, activity in non-dividing cells, and bystander activity. A dose-escalation study (ClinicalTrials.gov: NCT03451162) in patients with HER2-positive metastatic breast cancer, with the primary objective of evaluating the safety and tolerability of DHES0815A and secondary objectives of characterizing the pharmacokinetics, objective response rate, duration of response, and formation of anti-DHES0815A antibodies, is reported herein. Despite early signs of anti-tumor activity, patients at higher doses develop persistent, non-resolvable dermal, ocular, and pulmonary toxicities, which led to early termination of the phase 1 trial.

Amplification of the HER2/*erbB2* (human epidermal growth factor receptor 2) gene is predictive of higher probability of disease recurrence and poor overall survival in patients with breast cancer[1]. These observations, together with cell surface accessibility of the HER2 extracellular domain (ECD), resulted in development of antibody-based therapies directed against HER2. Trastuzumab, the first approved HER2 antibody, binds subdomain IV of the HER2 ECD and has multiple mechanisms of action (MOA), including inhibition of growth-factor-independent signaling through HER2-HER3-PI3K, inhibition of HER2 ECD shedding, and recruitment of immune effector cells via the Fc region to mediate ADCC (antibody-dependent cellular cytotoxicity)[2,3]. Trastuzumab was approved for use in combination with chemotherapy in first-line HER2-positive metastatic breast cancer

(HER2+ mBC)[4], followed by approval in the adjuvant setting[5], and is widely used with other systemic agents, such as chemotherapy, in later lines of treatment. In contrast to trastuzumab, pertuzumab binds subdomain II (dimerization domain)[6] and thereby inhibits growth factor mediated signaling by disrupting HER2 association with other HER/ErbB receptors[3]. In addition, pertuzumab mediates ADCC similar to trastuzumab[7]. Results to date suggest that engagement of HER2 at multiple epitopes confers clinical benefit. As such, pertuzumab is given in combination with trastuzumab and chemotherapy in the neoadjuvant[8], adjuvant[9,10] and first-line setting[10] in HER2+ BC patients.

HER2-directed antibody-drug conjugates (ADCs) provide an additional modality for treating HER2+ disease by selective delivery of cytotoxic agents to tumor cells. Trastuzumab emtansine (T-DM1), the

✉e-mail: gdl@gene.com

first HER2 ADC approved, is comprised of trastuzumab linked to the anti-microtubule agent DM1 through the uncleavable MCC linker[11]. T-DM1 is indicated for use in HER2+ mBC patients after treatment with trastuzumab and a taxane[12], as well as in the adjuvant setting in patients with residual disease following neoadjuvant therapy[13]. The success of T-DM1 as the first antibody-drug conjugate approved in solid tumors led to interest in developing other ADCs for HER2+ disease. The majority of these efforts were ultimately discontinued for undisclosed reasons. More recently, trastuzumab deruxtecan (T-DXd) received approval for treatment of HER2+ mBC[14,15] as well as approval in HER2+ metastatic gastric cancer[16]. Like T-DM1, T-DXd utilizes trastuzumab as the antibody backbone. However, in T-DXd, trastuzumab is linked via a cleavable peptide linker to the topoisomerase I inhibitor DXd, a derivative of SN-38[17].

Although impressive improvements have been made in overall survival (OS) in the adjuvant setting with trastuzumab/pertuzumab/ chemotherapy[9] and T-DM1[13], metastatic disease remains incurable. Because the treatment of mBC is palliative rather than curative in intent, improvement in survival is an important treatment goal. The current treatment landscape for mBC is evolving, with investigational medicines in development and recent new approvals[18].

Development of DHES0815A was initiated shortly after the approval of T-DM1 to potentially offer a different treatment option for HER2+ malignancies. DHES0815A is comprised of a HER2 THIOMAB[19] antibody, humanized 7C2 (hu7C2), that binds subdomain I of the HER2 ECD, linked to a modified PBD (pyrrolobenzodiazepine) dimer, PBD-monoamide, via a hindered disulfide linker conjugated at a cysteine engineered into light chain site lysine 149 (LC K149C; Kabat and EU numbering) optimized for enhanced stability[20]. In nonclinical models, DHES0815A demonstrates anti-tumor efficacy in HER2+ models of breast and gastric cancer, including models insensitive to T-DM1. These non-clinical data present a viable rationale to explore the pharmacokinetics, safety, and tolerability for preliminary assessment of anti-tumor activity of DHES0815A in patients with HER2+ mBC. Accordingly, a first-in-human phase 1 clinical study was conducted to evaluate DHES0815A as a single agent in patients with advanced and/or metastatic HER2+ BC for whom established treatment had proven ineffective.

The data herein describe design of DHES0815A, mechanisms of action, efficacy in HER2+ and HER2-low xenograft models, safety studies in non-human primates, and phase 1 safety and efficacy data. In this work we show that, despite strong and compelling preclinical data and early signs of anti-tumor activity in patients, a number of persistent safety findings resulted in termination of the phase 1 study.

## Results

### Design rationale for DHES0815A

DHES0815A was developed with the intent to design an ADC with a different antibody, linker and cytotoxic agent than T-DM1. Given that microtubule inhibitor (e.g., taxane)-based regimens with trastuzumab and pertuzumab are commonly used in the treatment of HER2+ breast cancer, targeting a unique HER2 epitope and delivering a DNA damaging cytotoxic agent were prioritized, allowing potential combination with, as well as minimizing cross-resistance to current treatment regimens.

The antibody selected, 7C2, binds subdomain I of the HER2 ECD. This epitope differentiates 7C2 from trastuzumab (subdomain IV) and pertuzumab (subdomain II), thus allowing for potential combination with existing HER2 antibody therapeutics. The binding affinity ($K_D$) of humanized 7C2 (hu 7C2) for human HER2 and cynomolgus monkey HER2 are 0.8 nM and 0.57 nM, respectively, as assessed by Scatchard analysis (Supplementary Fig. 1). Like trastuzumab and pertuzumab[21], hu 7C2 does not bind rodent neu (Supplementary Fig. 1). FACS analysis in HER2+ BC cells demonstrates full binding of 7C2 in the presence of trastuzumab, pertuzumab or the combination (Supplementary Fig. 2). Humanized 7C2 was engineered for conjugation at cysteine

substitutions on the light chain at lysine149 using THIOMAB technology[22]. Hu7C2 THIOMAB LC K149C is designated MHES0488A. The two light chain (LC) lysine-to-cysteine mutations result in homogeneous conjugation to DAR 2 (drug:antibody ratio of 2). The linker selected is a self-immolative hindered disulfide linker, previously demonstrated to provide better safety with PBD dimer payloads, while retaining efficacy[20]. Both conjugation site and linker were optimized for enhanced in vitro and in vivo stability.

Cytotoxic agents with mechanisms different than DM1, primarily DNA damaging agents, were investigated as potential payloads for this ADC, with a focus on PBD dimers. Potency of the payload was key, in terms of ADC exposure, as clinical data with T-DM1 demonstrates linear pharmacokinetics (PK) at doses ≥ 2.5 mg/kg, and rapid clearance at doses below[23]. Our prior experience with PBD dimer ADCs demonstrated that MED (minimum efficacious dose in mouse xenograft studies) and MTD (maximum tolerated dose in cynomolgus monkey) were 1 mg/kg or lower. It was thus concluded that PBD ADCs would have a narrow therapeutic index (TI) and would not allow dosing in the linear PK range. In order to provide optimal PK, as well as improve the therapeutic index, modifications were made to reduce the potency of the PBD dimer SG2057[24].

PBD dimers are highly cytotoxic agents that bis-alkylate DNA guanine residues, resulting in DNA minor groove inter-strand crosslinks[24]. The imine pharmacophores are the DNA reactive moieties in SG2057 (Fig. 1a). SG2057 was chemically modified to monoamine or monoamide derivatives to render one imine unreactive, thus resulting in conversion to DNA mono-alkylating compounds, which alkylate but do not cross-link DNA. DNA binding assays were performed using double-stranded oligonucleotides with one guanine (G), the site of alkylation, per strand. The bis-alkylator PBD dimer forms adducts with the oligonucleotide by alkylating G in both strands, whereas a monoalkylator reacts with one G. Under these experimental conditions, 99-100% of spiked oligonucleotide was converted to PBD adducts after incubation with the parent PBD dimer and slightly lower alkylation of 94% with the PBD-monoamine. In contrast, 57% of the oligonucleotide was converted to adducts with the PBD-monoamide, in alignment with our goal (Fig. 1b). Assessment of in vitro activity across large cancer cell line panels (Fig. 1a table) showed that the PBD-monoamine retained potency (average $IC_{50}$ 8.6 nM) relative to the parent PBD (average $IC_{50}$ 1.6 nM), whereas potency of the monoamide was reduced (average $IC_{50}$ 27.1 nM), consistent with the DNA binding data. A similar potency trend for PBD vs. PBD-monoamine vs. PBD-monoamide was observed with 7C2 disulfide-conjugated PBDs (see Supplementary Fig. 3 for ADC structures), with reduced potency of the 7C2 disulfide-PBD-monoamide compared to the conjugated monoamine or PBD dimer (Fig. 1c, left panel). Importantly, the goal of reducing the potency of PBD payloads to achieve higher dosing was confirmed in vivo (Fig. 1c, middle panel). In the MMTV-HER2 Fo5 model, dose-dependent tumor growth inhibition was observed with 5 and 10 mg/kg 7C2-disulfide PBD-monoamide. Tumor stasis out to day 25 was achieved with 15 mg/kg following a single IV administration. In contrast, treatment with 0.5 mg/kg conjugated PBD dimer resulted in stasis, with tumor regression at 1 mg/kg. Two additional PBD mono-alkylators, olefin and morpholine derivatives, underwent similar assessment, but did not demonstrate improved properties compared to the PBD-monoamide (Supplementary Fig. 4). Taken together, these results led to the selection of disulfide-linked PBD-monoamide for further development. The structure of hu7C2 THIOMAB LC K149C-disulfide PBD-monoamide is shown in Fig. 1d; this clinical candidate molecule is designated DHES0815A.

### In vitro characterization and mechanisms of action of DHES0815A

As intracellular delivery of cytotoxic agents is the rationale for ADC activity, we first confirmed internalization of DHES0815A into HER2+

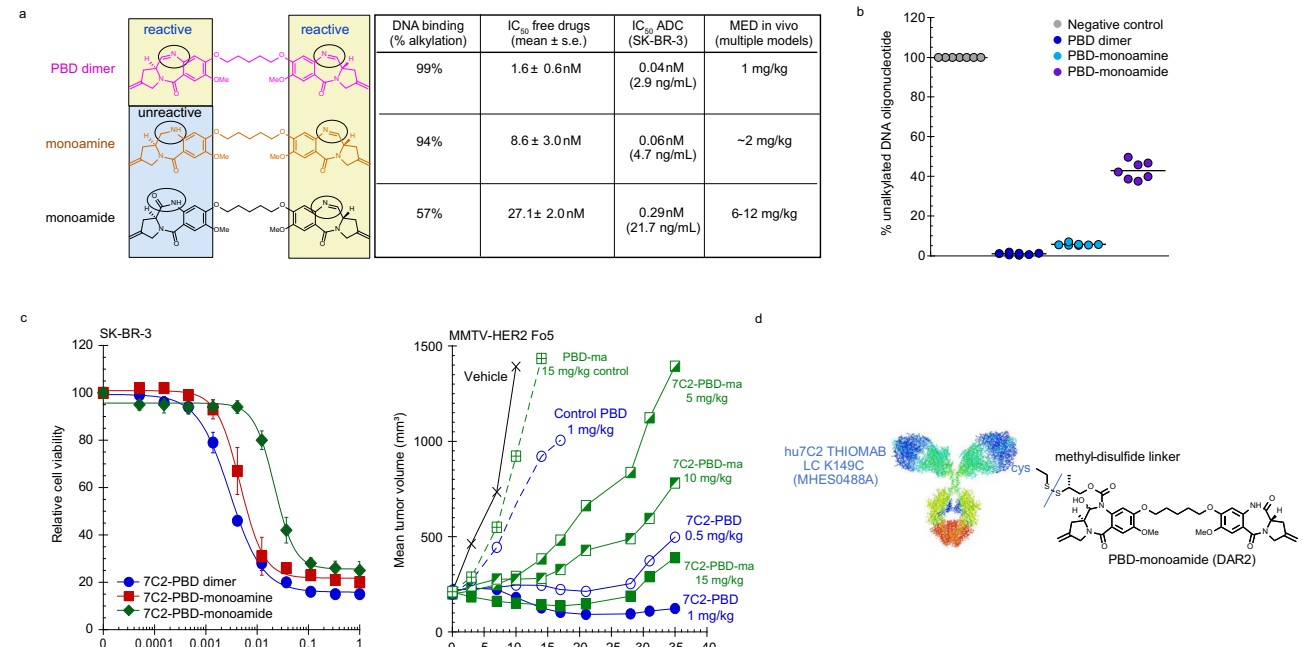

**Fig. 1 | Design and characterization of PBD derivatives. a** Structures of PBD, PBD-monoamine and PBD-monoamide, and summary table for DNA binding and drug activity (free drugs and ADCs in vitro and in vivo). In the left panel, circles denote reactive imine moiety and changes thereof to generate monoamine and monoamide derivatives. For the summary table: large cell line screens were performed to assess potency of the PBD dimer, PBD-monoamine and -monoamide. The PBD dimer was tested in 7 different cell line screens; the PBD-monoamide in 3 screens; and the PBD-monoamine in 2 screens ($n = 3$ wells per treatment group for all). The number of cell lines tested per screen ranged from 50-643 for the PBD; 73-146 for the PBD-monoamide and 72-147 for the PBD-monoamine. Data shown are the pooled mean $IC_{50}$ values ± standard error from the respective screens. ADC $IC_{50}$ was determined in SK-BR-3 cells (data are mean $IC_{50}$ for 3 independent experiments, $n = 4$ wells per treatment group for each; see Fig. 1c for graph). DNA alkylation percent is derived from data shown in part 1b. For determining MED, multiple xenograft models were used, with $n = 5$ to $n = 10$ mice per group. MED = minimum efficacious dose, defined as a single injection dose that results in tumor stasis at day 21 in mouse xenograft models. **b** DNA binding of PBD dimer, PBD-monoamine and PBD-monoamide as assessed by alkylation of double-stranded DNA oligonucleotides (each point represents oligonucleotide reactions, from 2 separate experiments). **c** Activity of DHES0815A compared to conjugated PBD dimer or PBD-monoamine (same antibody and linker) in SK-BR-3 cells in vitro and in MMTV-HER2 Fo5 model in vivo (PBD and PBD-monoamide conjugates only, $n = 8$ mice per group). **d** Structure of DHES0815A, comprised of 7C2 THIOMAB (MHES0448A), methyl-disulfide linker and PBD-monoamide. Source data are provided as a Source Data file.

breast cancer cells. Immunofluorescence microscopy demonstrated that both DHES0815A (ADC) and MHES0488A (antibody) accumulated in late endosomes and lysosomes (Supplementary Fig. 5); indicating that MHES0815A internalizes and traffics as expected, and that conjugation with PBD-monoamide did not affect antibody uptake or trafficking. In vitro cell killing assays demonstrated selectivity of DHES0815A for HER2+ BC cells (SK-BR-3), with no activity on HER2-negative (HER2-) MCF7 (Fig. 2a) or MDA-MB-468 cells (Supplementary Fig. 6a). Both MCF7 and MDA-MB-468 are sensitive to unconjugated PBD-monoamide treatment (Supplementary Fig. 6b). Additionally, a non-targeted (CD22) disulfide PBD-monoamide ADC showed no activity on the HER2+ or HER2- cells, indicating optimal linker stability and low non-specific uptake. The unconjugated antibody MHES0488A also showed no activity, differentiating this antibody from trastuzumab and pertuzumab, which inhibit HER2-mediated signaling and cell growth. HER2+ BC cells showed differential responses to DHES0815A vs. T-DM1 (Fig. 2a, right panels). Response to the two ADCs correlated with intrinsic sensitivity to the two different payloads (Supplementary Fig. 7) and was not due to differences in target expression (all cell lines tested were HER2 IHC 3 + ) or expression of drug efflux pumps such as Pgp (cells were negative for Pgp expression). The differential response to DHES0815A may represent potential expansion of the treated patient population. Based on the difference in payload MOA, we anticipated that DHES0815A would also be differentiated from T-DM1 by affecting non-dividing cells. To assess actively proliferating vs. quiescent cells, two models of growth arrest/quiescence were utilized. Normal human epidermal keratinocytes were growth arrested by

culturing the cells to confluence to induce contact inhibition. Serum deprivation was used to arrest SK-BR-3 cells[25]. Both PBD-monoamide and S-methyl DM1 (cell permeable version of DM1) reduced cell viability in dividing keratinocytes and SK-BR-3 cells (Supplementary Fig. 8). As expected of a tubulin binding agent, S-methyl DM1 was inactive on non-dividing cells. In contrast, treatment with PBD-monoamide resulted in cell viability reduction in non-dividing cells (albeit to a lesser extent than dividing cells). Finally, activity of both DHES0815A and MHES0488A was assessed on normal human cells (Supplementary Fig. 9), which do not overexpress HER2. MHES0488A did not affect growth of human mammary epithelial cells, umbilical vein endothelial cells, small airway epithelial cells or renal proximal tubule epithelial cells. DHES0815A inhibited growth only at the highest concentrations tested (>1 µg/mL), as observed with other ADCs due to non-specific uptake[11].

ADCC is a mechanism of tumor cell killing for antibodies with wildtype Fc. Hu7C2 mediated ADCC in SK-BR-3 and KPL-4 cells in the presence of PBMCs (peripheral blood mononuclear cells). ADCC was not affected by conjugation with MCC-DM1 or with disulfide-PBD-monoamide (Fig. 2b). Induction of ADCC was then compared to T-DM1. Interestingly, T-DM1-mediated cell killing in the presence of PBMCs was slightly better than 7C2-DM1 or DHES0815A, likely due to closer proximity of the trastuzumab binding site (subdomain IV) to the cell membrane, facilitating closer contact of immune cells with tumor cells.

In recent years, it has been appreciated that bystander activity of ADCs may be a desirable attribute, depending on target as well as ADC

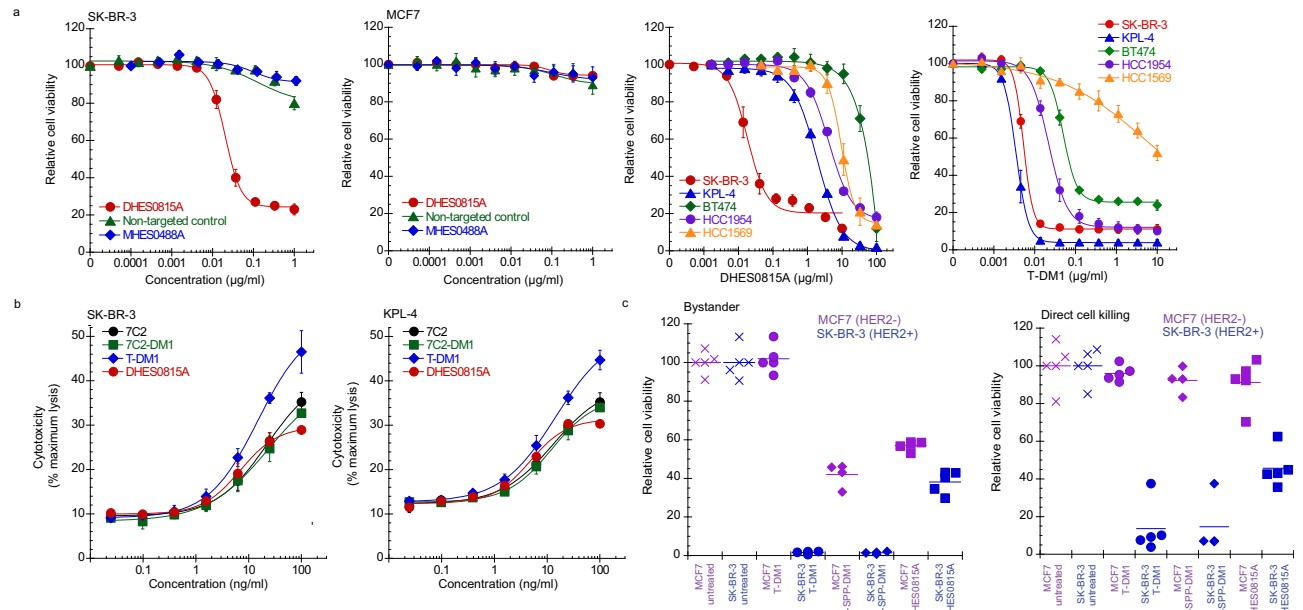

**Fig. 2 | In vitro characterization of DHES0815A. a** DHES0815A activity is HER2-dependent and is differentiated from T-DM1 ($n = 4$ individual wells per treatment group, data points are mean ± standard error of the mean pooled from 3 independent experiments). **b** DHES0815A induces ADCC in HER2+ SK-BR-3 and KPL-4 cells in the presence of PBMCs ($n = 3$ individual wells per treatment group; data points are mean of individual replicates ± standard error of the mean from 2 independent experiments). **c** DHES0815A mediates bystander activity, as indicated by killing of HER2- MCF7/NucLight Red in the presence of HER2+ SK-BR-3/H2B-GFP and ADCs with reducible linkers (T-SPP-DM1, DHES0815A). Data points are from 3 separate experiments. Source data are provided as a Source Data file.

linker-drug. ADCs with cell permeable payloads linked with cleavable linkers can elicit killing of neighboring tumor cells following ADC internalization, trafficking to lysosomes, linker cleavage and exit of the payload from the target cell[26–28]. This bystander activity is likely important in tumors with heterogeneous target expression. DHES0815A has a reducible disulfide linker and therefore was hypothesized to elicit bystander activity. To test this, co-cultures of HER2-MCF7/Nuclight Red and HER2+ SK-BR-3/H2B-GFP were treated with DHES0815A, trastuzumab-SPP-DM1[11] (ADCs with reducible disulfide linkers) or T-DM1 (uncleavable linker), and the number of live cells assessed by FACS (Fig. 2c). As expected, due to the uncleavable linker, treatment with T-DM1 did not result in killing of HER2- MCF7 cells in co-culture with SK-BR-3. In contrast, both disulfide-linked ADCs elicited bystander activity, as demonstrated by reduction of MCF7 cell number. Lack of direct activity of ADCs on target-negative MCF7 cells was confirmed by treatment in the absence of SK-BR-3 cells. These data validate that the disulfide linker and cell permeability of PBD-monoamide enable DHES0815A to induce bystander activity.

As DHES0815A is comprised of a payload with a different MOA than T-DM1, studies were undertaken to compare DHES0815A to T-DM1 for induction of apoptosis and DNA damage response markers. Kinetics of caspase 3 activation in SK-BR-3 cells were assessed using Incucyte (Fig. 3a). Caspase 3 activity was induced by T-DM1 after 1 day, with maximal activation at 2.5 days. In contrast, DHES0815A displayed delayed apoptosis induction, with caspase activation starting at 2.5 days and peaking at day 5. This slower rate of caspase 3 activation reflects the different nature of a DNA active vs. a tubulin binding payload. Lysosomal processing of both ADCs releases free drug into the cytosol. However, the released PBD-monoamide must then cross the nuclear membrane, alkylate DNA and induce DNA damage. The extent of apoptosis induction with DHES0815A was, however, greater than with T-DM1. Unconjugated antibodies (hu7C2, trastuzumab) showed no increase in caspase 3 activity compared to untreated controls. Time-dependent induction of PARP cleavage in SK-BR-3 cells was similar to caspase 3 activation, with cleaved PARP fragment appearing at an earlier time point with T-DM1 vs. DHES0815A (Fig. 3b). Because combination of

DHES0815A with trastuzumab/pertuzumab was part of the planned clinical development strategy, apoptosis was also investigated in cells treated with these combinations. Simultaneous antibody engagement of different receptor epitopes is reported to enhance internalization, a desirable outcome for ADC treatment[29,30]. Internalization of HER2 was assessed by immunofluorescence upon treatment of SK-BR-3 cells with DHES0815A alone or combined with trastuzumab, pertuzumab or the combination. Disappearance of HER2 from the cell membrane and localization into lysosomes was most pronounced with the triple combination (Supplementary Fig. 10). Accordingly, apoptosis induction by the triple combination of DHES0815A, trastuzumab and pertuzumab was greater than single agent or double combination treatment (Supplementary Fig. 10, right panel), thus supporting our hypothesis that enhanced activity can be achieved with DHES0815A in the presence of therapeutic HER2 antibodies.

Markers of DNA damage were evaluated with both free PBD-monoamide and DHES0815A in SK-BR-3 cells (Fig. 3c). Phospho-H2AX, phospho-p53, total p53 and phospho-CHK2 were upregulated to similar levels with 100 ng/mL DHES0815A and 1 nM PBD-monoamide (the payload equivalent concentration for DHES0815A). Time-dependent regulation of DNA damage and mitotic markers was compared for DHES0815A and T-DM1. DHES0815A, but not T-DM1, induced total and phospho-p53 in a time-dependent manner (Fig. 3d). Phospho-H2AX was induced by DHES0815A and by T-DM1 as well. Activation of H2AX by T-DM1, a microtubule inhibitor, was unexpected, but consistent with reports that H2AX can be phosphorylated by the DNA-PK/CHK2 pathway in cells undergoing mitotic arrest[31]. The mitotic marker histone H3 was phosphorylated by T-DM1, which arrests cells in G2/M[11], but not by DHES0815A, which causes predominantly S-phase arrest (Supplementary Fig. 11).

## In vivo efficacy of DHES0815A across breast and gastric cancer models

Efficacy of DHES0815A, as a single agent and combined with standard-of-care (SOC) therapies, was evaluated in cell line and PDX (patient-derived xenograft) models. Dose-dependent tumor growth inhibition

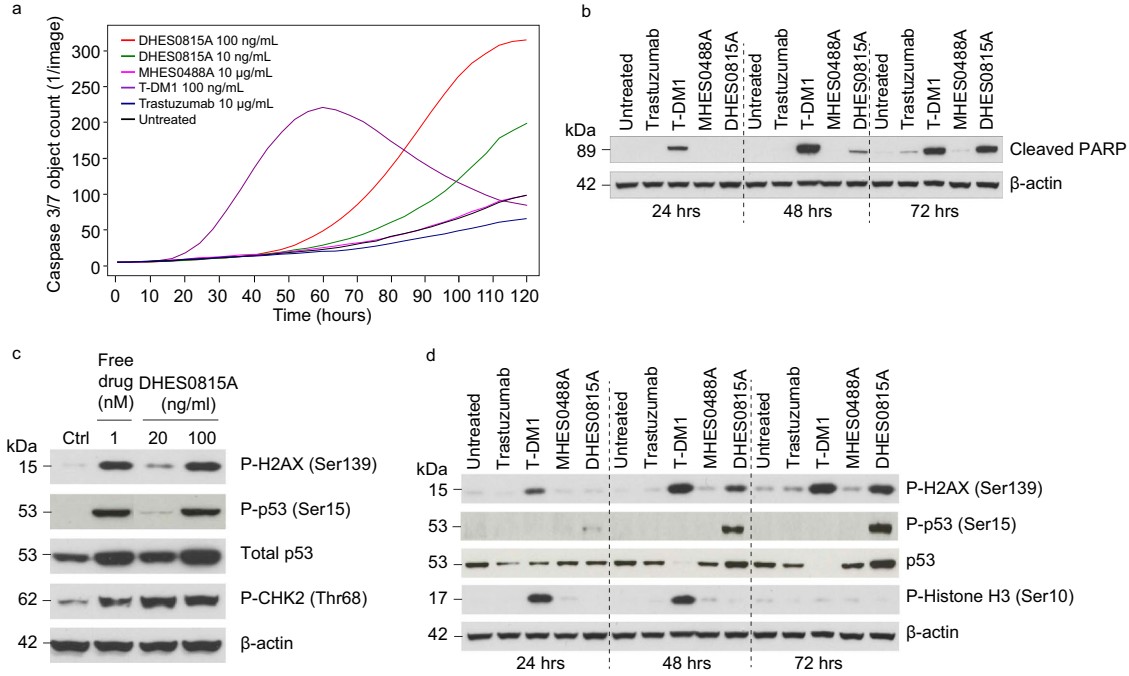

**Fig. 3 | DHES0815A induces apoptosis and markers of DNA damage in SK-BR-3 cells. a** Kinetics of caspase 3 activation for DHES0815A compared to T-DM1, MHES0448A and trastuzumab (n = 6 individual wells for each treatment group, the study was repeated 4 times). **b** Time-dependent PARP cleavage after treatment with antibodies (trastuzumab, MHES0488 A) or ADCs (T-DM1 or DHES0815A). **c** Induction of DNA damage markers (phospho-H2AX, phospho-/total p53 and phospho-CHK2) with free PBD-monoamide and DHES0815A. **d** Time-dependent induction of DNA damage and mitotic (phospho-histone H3) markers after treatment with unconjugated antibodies and ADCs. Western blot studies in (**b**)–(**d**), were performed 2 times with the same results. Source data are provided in Source Data file.

was demonstrated in the HER2+ (IHC 3+ ) breast cancer models MMTV-HER2 Fo5, HCC1569 X2 cell line and WHIM8 PDX model (Fig. 4a), as well as in HER2+ (IHC 3+ ) gastric cancer PDX models STO410 and STO41 (Fig. 4b, see Supplementary Fig. 12 for HER2 IHC for these models). In the PDX models WHIM8, STO410 and STO41, DHES0815A was more efficacious than T-DM1 administered at the same doses (5, 10 or 15 mg/kg), demonstrating that although DHES0815A has less pay-load (DAR 2), it is more potent than T-DM1 (DAR 3.5). DHES0815A was also compared to T-DM1 in four HER2-low (HER2 IHC 1+ or IHC 2+ /ISH-) breast PDX models (Fig. 4c, see Supplementary Fig. 12 for HER2 IHC). T-DM1 showed little or no activity at 7 mg/kg (equivalent to the clinical dose of 3.6 mg/kg), as expected due to the uncleavable linker and cell impermeable payload which limit bystander activity, which is proposed to play a role in ADC efficacy in HER2-low or HER2 hetero-geneous tumors[17]. In contrast, DHES0815A induced tumor stasis or regression at the same dose, with no further improvement of anti-tumor activity at 15 mg/kg. Combination therapy in HER2+ MMTV-HER2 Fo5 and HCC1569 X2 models with DHES0815A and SOC agents T-DM1 or docetaxel resulted in enhanced combination activity vs. single agent treatment (Fig. 4d). Because DHES0815A was active in HER2+ and HER2-low xenograft models, it was important to demon-strate target (HER2)-dependent activity. DHES0815A was compared to a CD22 ADC with the same linker-payload in the CD22-expressing lymphoma model WSU-DLCL2 (HER2-). Dose-dependent activity was demonstrated for the CD22 ADC, with no observed efficacy of DHES0815A (Supplementary Fig. 13), indicating no target-independent activity for DHES0815A.

## Comparison of DHES0815A with trastuzumab deruxtecan (T-DXd)
Because T-DXd was undergoing clinical testing during preclinical development of DHES0815A, activity of the two ADCs was assessed in HER2+ BC models. First, a comparison of free drug potency was per-formed in 2 HER2+ cell lines (SK-BR-3 and KPL-4) across different ADC payloads (Fig. 5a). The PBD dimer and S-methyl-DM1 were the most potent of the free drugs tested. The reduced potency PBD-monoamide was more potent than the topoisomerase 1 inhibitors SN-38 and its derivative DXd, payloads on the ADCs sacituzumab govitecan and T-DXd, respectively[17,32]. Low potency of SN-38 and DXd likely explains the rationale for high DAR (-8) of these ADCs. The T-DM1 refractory PDX model WHIM8 was selected to compare potency of DHES0815A and T-DXd (Fig. 5b). A single 5 mg/kg dose of DHES0815A resulted in modest transient tumor regression, while the same dose of T-DXd resulted in sustained tumor regression in all animals. To address the effect of different DARs on efficacy (DAR 1.8 for DHES0815A and DAR 7.9 for T-DXd), a drug-normalized dose of 18 mg/kg DHES0815A was tested, which showed similar efficacy to the drug-matched 5 mg/kg dose of T-DXd (tumor regression in all animals). As DHES0815A and T-DXd employ different antibody backbones, hu7C2-DXd was also evaluated. Efficacy with hu7C2-DXd was identical to T-DXd, indicating that the 7C2 antibody does not alter efficacy compared to the same ADC with trastuzumab. Overall, these data demonstrate similar effi-cacy between DHES0815A and T-DXd.

## Safety studies in nonhuman primates
Cynomolgus monkey (*Macaca fascicularis*) is an appropriate model for safety assessment of target-dependent toxicities of DHES0815A as hu7C2 binds to cynomolgus monkey erbB2 with similar affinity as human. Preliminary safety studies investigated DHES0815A doses between 1 and 24 mg/kg Q3W (every 3 weeks) X 2, with a 3 week recovery period. With this dosing regimen, the MTD (maximum tol-erated dose) was determined to be greater than 24 mg/kg. A sub-sequent study implemented a bridging dose of 16 mg/kg, as well as 24 and 36 mg/kg with extended dosing of Q3W X 4 (3 week recovery).

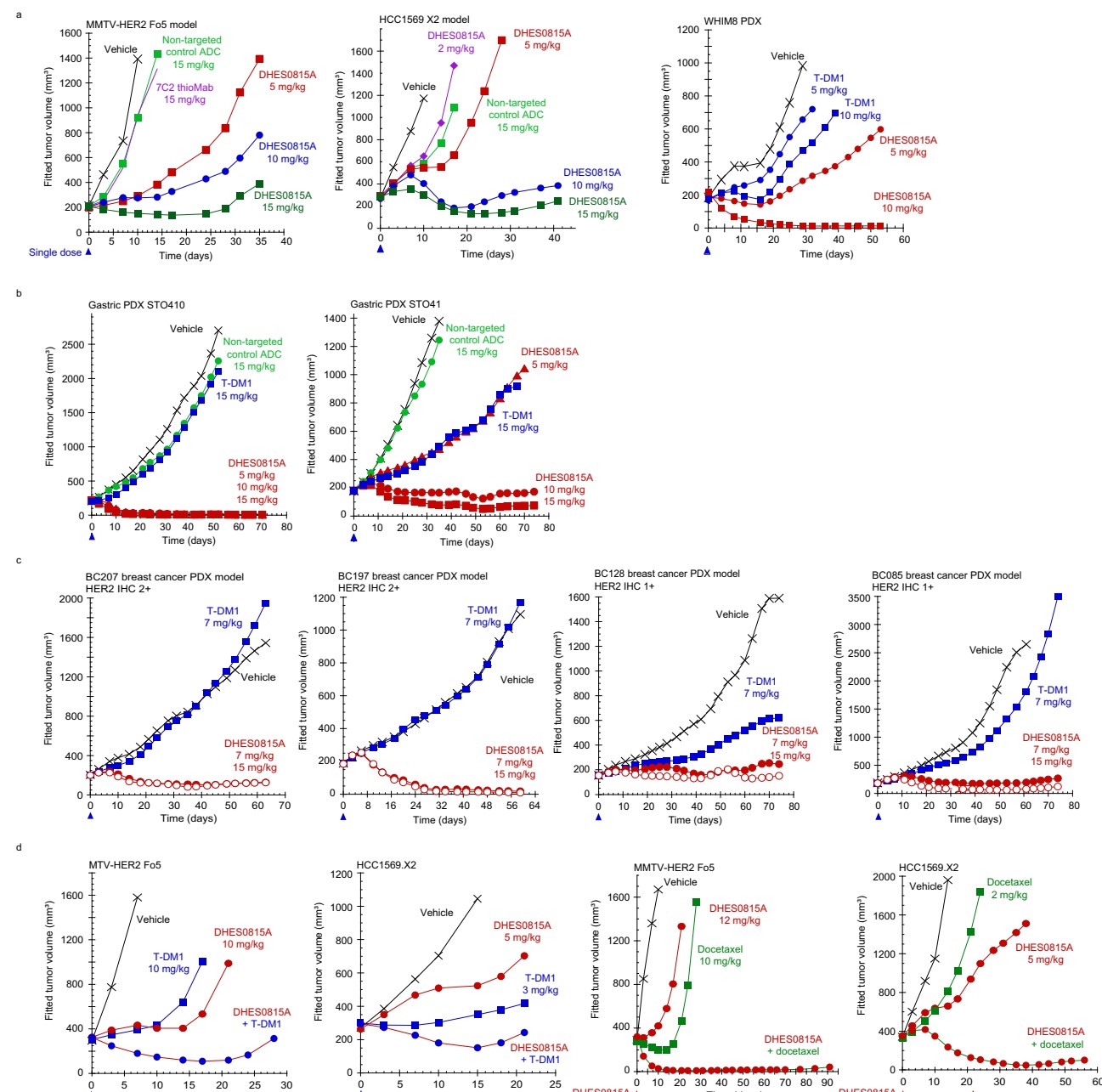

**Fig. 4 | In vivo efficacy of DHES0815A in HER2+ breast and gastric cancer models, HER2-low breast PDX models, and combinability with standard-of-care therapies. a** Dose-dependent tumor growth inhibition after single dose DHES0815A treatment in HER2+ MMTV-HER2 Fo5 (*n* = 8 per group) and HCC1569 X2 (*n* = 5 mice per group), and comparison to T-DM1 in the WHIM 8 PDX model (*n* = 10 per group). **b** Efficacy of DHES0815A compared to T-DM1 in HER2+ STO410 and STO41 gastric cancer PDX models (*n* = 10 mice per group). **c** Efficacy of

DHES0815A vs. T-DM1 in HER2-low (IHC 1+ or 2+ /ISH-) breast cancer PDX models BC207, BC197, BC128 and BC085 (*n* = 10 mice per group for all studies). **d** Enhanced efficacy of DHES0815A combined with T-DM1 or docetaxel compared to single agent activity in MMTV-Fo5 (*n* = 5 mice per group for T-DM1 combination; *n* = 7 mice per group for docetaxel combination) and HCC1569 X2 (*n* = 8 mice per group for T-DM1 combination; *n* = 8 mice per group for docetaxel combination) models. Source data are provided as a Source Data file.

None of the doses were tolerated, with the MTD < 16 mg/kg, indicating the importance of repeat dosing (> 2) for more rigorous safety assessment. The final GLP toxicology study design implemented 4, 8, and 12 mg/kg Q3W X 5, with a 7-week recovery period. As delayed toxicities are common with agents that induce DNA damage, a prolonged recovery period is essential. Clinical and anatomic findings are described in Supplementary Table 1. Hematology findings included mild decreases in lymphocytes and eosinophils at all 3 doses, with mild decreased reticulocyte count at the highest dose (12 mg/kg). Reversible lymphoid depletion (minimal) was observed only at 12 mg/kg. Skin hyperpigmentation occurred at all doses but was irreversible only

at 12 mg/kg. Slight corneal pigmentation was observed with the 8 and 12 mg/kg doses. Increases in lung alveolar macrophages were observed at the highest dose only, with 1 animal showing focal alveolar degeneration. Although these findings were reversible as they were not observed in the recovery animals, it is possible that the focal alveolar degeneration is an infrequent finding that might be more prevalent, and potentially not reversible, with larger numbers of animals. The observed safety signals were considered target-independent, as similar safety findings were observed for other ADCs, including non-targeted ADCs, with PBD-type payloads. Taken together, the HNSTD (highest non-severely toxic dose) was determined to be 12 mg/kg. MTD was not

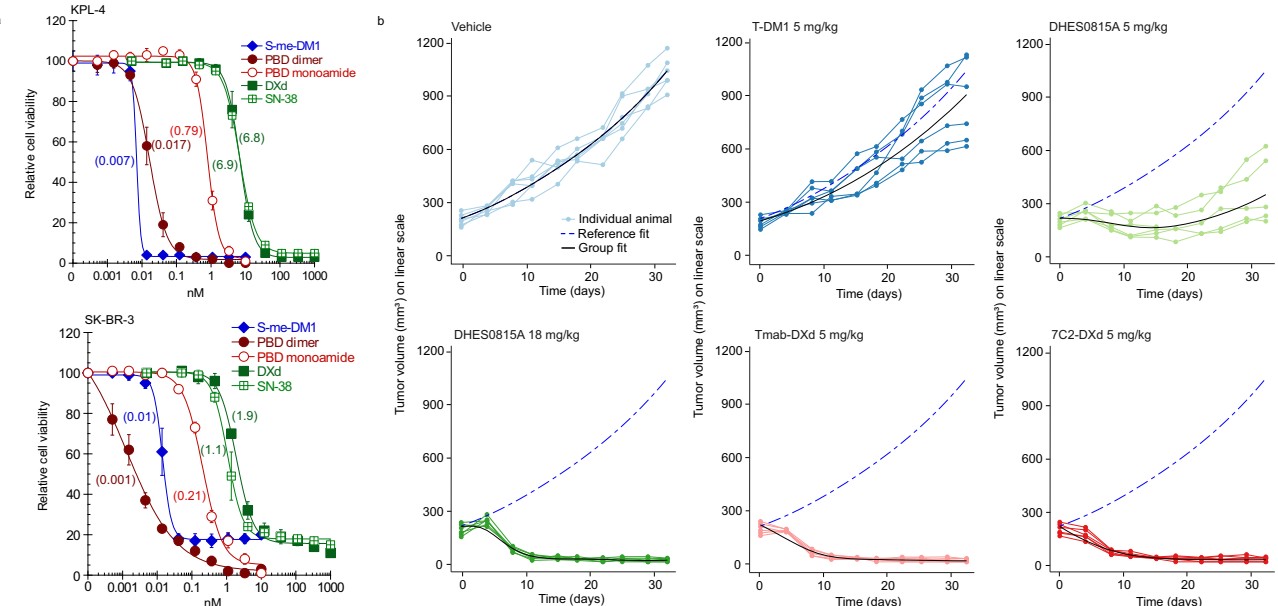

**Fig. 5 | Comparison of free drug and ADC activity for DHES0815A and trastuzumab deruxtecan (T-DXd). a** In vitro potency of different ADC payloads (DM1, PBD dimer, PBD-monoamide, DXd and SN38) in KPL-4 and SK-BR-3 cells ($n = 4$ individual wells per data point; data are represented as mean ± standard error from 3 independent experiments). **b** In vivo efficacy of DHES0815A compared to T-DXd in the WHIM 8 HER2+ PDX model ($n = 6$ mice per group). Source data are provided as a Source Data file.

reached in this study. Pharmacokinetics of DHES0815A showed dose-proportional exposure (Supplementary Fig. 14), with linear PK above 4 mg/kg.

Therapeutic index is determined using the MTD (or HNSTD) from nonhuman primate (NHP) safety studies and MED (minimal efficacious dose) from mouse xenograft studies. TI comparison across different molecules is complicated by 1) different definitions of MED (dose that results in any level of tumor growth inhibition vs. dose that results in tumor stasis) and 2) different dosing regimens used in NHP studies. There are several approaches for estimating TI, including the more rigorous determination of TI based on exposure or using body surface area (to account for differences in rodent vs. NHP) vs. simply using MTD and MED based on dose (mg/kg). For TI calculation in our models, we define MED as a single dose that results in tumor stasis for at least 21 days. HNSTD was derived from our safety study with a dosing regimen of Q3W X 5. Given these parameters, the exposure-based (total antibody AUC) TI for DHES0815A is 2-4, and the body surface area-based (mg/m²) TI is 4-8. The range reflects using MED from multiple xenograft studies with a range of sensitivity to DHES0815A. In contrast, there is no TI for HER2 ADCs (evaluated at Genentech) with the bis-alkylator PBD payloads, as MED values were ≤ 1 mg/kg and MTD values < 1 mg/kg.

## Phase 1 efficacy, PK and safety data for DHES0815A

The primary objective of the phase 1 open-label, dose-escalation study (NCT03451162) utilizing a 3 + 3 design was to evaluate safety and tolerability of DHES0815A in patients with advanced or metastatic HER2+ BC. Secondary objectives were characterization of DHES0815A pharmacokinetics, objective response rate, duration of response, and formation of anti-DHES0815A antibodies. Fourteen patients were enrolled at doses of 0.6 to 6 mg/kg. Demographics of the enrolled patients are shown in Supplementary Table 2. All patients were required to have HER2+ disease as defined by ASCO/CAP 2013 guidelines at the time of enrollment[33].

Out of 14 evaluable patients, one patient (7%) dosed at 1.2 mg/kg experienced a confirmed complete response after 6 cycles according to RECIST v1.1[34]. Two patients achieved a partial response (14%), eight patients (57%) showed a confirmed best response of stable disease, two had progressive disease (14%) and one patient was not considered evaluable by the investigator (7%). The median duration of treatment was 64 days (43–62), and the patient achieving complete response remained on study for 32 months. Duration of response was not evaluated due to the small number of patients with a treatment response. Figure 6a shows a waterfall plot of change in target lesion SLD (sum of longest diameters) for the different doses, HER2 and hormone receptor status, prior lines of therapy, number of cycles of DHES0815A and best response.

The PK assay strategy for DHES0815A included measurement of 3 key analytes: total antibody, antibody-conjugated PBD-monoamide and unconjugated PBD-monoamide (Fig. 6b). $C_{max}$ and AUC of the three analytes were dose-proportional, with $C_{max}$ achieved immediately at the end of infusion. No apparent accumulation in exposure was observed with repeat doses of DHES0815A. Minimal systemic exposure of unconjugated PBD-monoamide was observed, with 1000-fold lower $C_{max}$ compared to antibody-conjugated PBD-monoamide across all doses. Total antibody half-life ($t_{1/2}$) was 1.3–3.6 days for the lower doses (0.6–2.4 mg/kg) and 9–10 days for 4–6 mg/kg doses (Supplementary Table 3). Antibody-conjugated PBD-monoamide $t_{1/2}$ was 1.5–3.5 days for 0.6–2.4 mg/kg dose range and 7–8 days for 4–6 mg/kg. PBD-monoamide $t_{1/2}$ was 6–8 days across all doses. Both total antibody and antibody-conjugated PBD-monoamide showed nonlinear PK at 0.6, 1.2 and 2.4 mg/kg due to target-mediated drug disposition, as expected. Linearity was obtained with doses of 4.0 mg/kg and above (Fig. 6b). The incidence of treatment-emergent anti-DHES0815A antibodies was 14.3% (2 of 14 evaluable patients). No conclusions were drawn concerning the potential impact of ADAs on PK, efficacy, or safety due to the limited number of ADA-positive patients.

All 14 patients were safety evaluable, and all experienced at least one treatment emergent adverse event (TEAE). Summaries of AEs are provided in Table 1. No DLT (dose-limiting toxicity) criteria were met during dose escalation for any cohort and the MTD was not reached. However, the onset of several dermatologic, ocular, and pulmonary adverse events presented in later cycles beyond the DLT window,

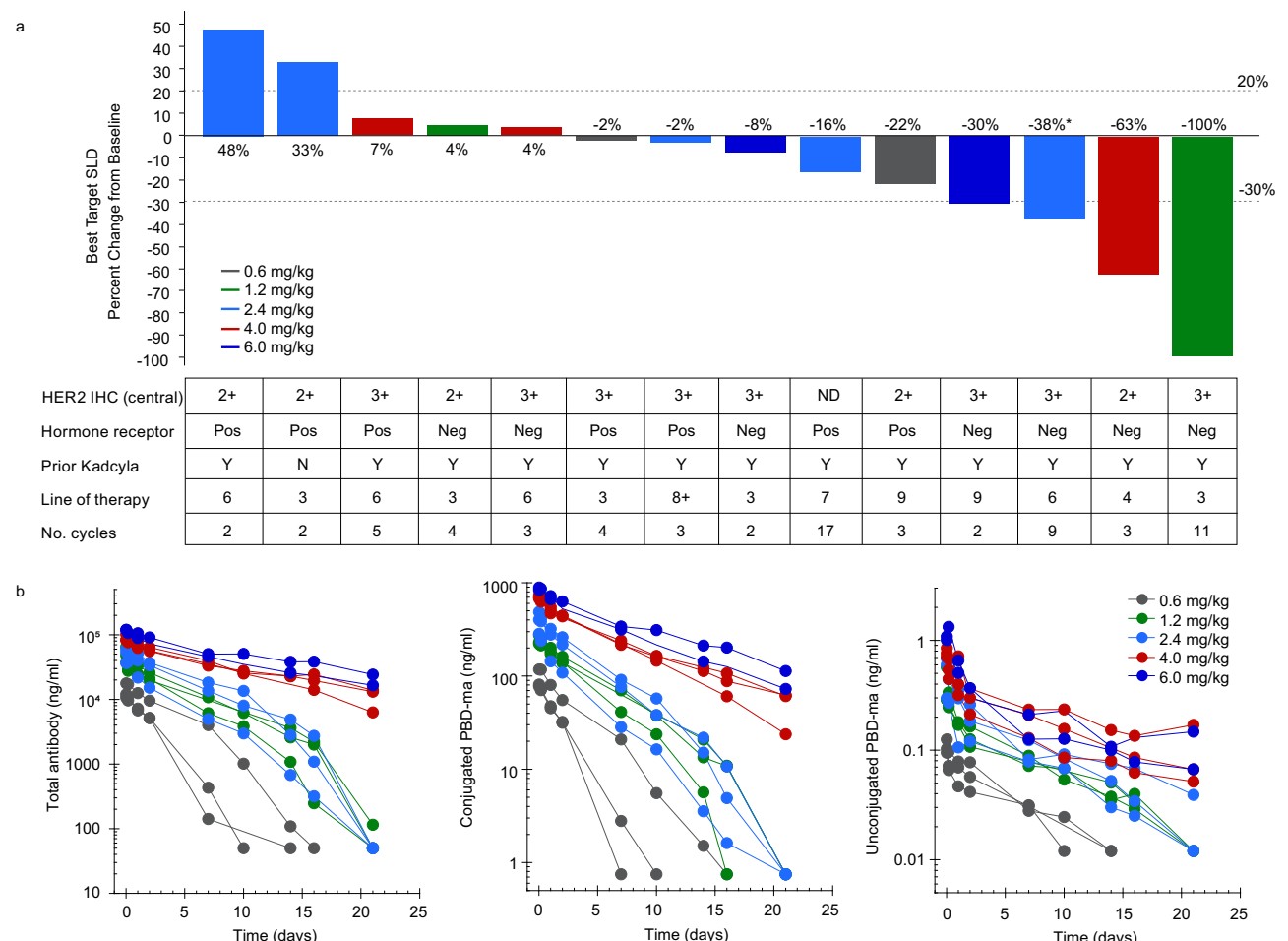

**Fig. 6 | Clinical activity and pharmacokinetics of DHES0815A. a** Waterfall plot for phase 1 dose-escalation, including patient histories. Doses shown are the highest administered (initially assigned dose for some patients, highest dose following intrapatient escalation for other patients). **b** Pharmacokinetic analysis from dose-escalation for total antibody, antibody-conjugated PBD-monoamide (acPBD-ma) and unconjugated PBD-monoamide. *PD due to new lesions. Source data are provided as a Source Data file.

limiting further dose escalation and eventually resulted in halting enrollment.

TEAEs reported in ≥ 20% of patients are shown in Table 1. Eleven patients (79%) experienced AEs considered related to DHES0815A; the most frequently reported (n ≥ 3 patients) included pruritus and rash (n = 5 each, 36%), fatigue (n = 4, 29%), skin hyperpigmentation, photophobia, and nausea (n = 3 each, 21%). Following three or more cycles at 4 and two or more cycles at 6 mg/kg, multiple safety events involving skin, eyes and lung emerged, often concurrently in a given patient; these events led to treatment discontinuation in all 5 patients treated at these doses. Dermatologic toxicities included pruritus, rash, and skin hyperpigmentation, and were managed with topical and oral antihistamines and corticosteroids. Ocular toxicities included photophobia, conjunctivitis, blepharitis, dry eye, eyelid/periorbital edema, punctate keratitis and eye pain. These toxicities manifested with clinical symptoms and were not identified with routine ophthalmic examinations that were required as part of the study. Management included antibiotics, ophthalmic lubricants, steroids, analgesics, topical anesthetics and other agents. Pneumonitis was reported in 2 patients dosed at or above 4 mg/kg and CT findings of ground glass opacities were reported for both patients. Management included systemic steroids. Due to the small number of patients, it was not clear if any medical intervention for the skin, eye, and lung toxicities improved the severity or duration of symptoms; all 5 patients had unresolved

sequelae from some of these toxicities at the time of discontinuation from study follow-up. Supplementary Fig. 15 depicts rash and pulmonary findings and outlines the time course of AEs in a patient who was treated at 4.0 mg/kg for 3 cycles. Two patients assigned to lower doses (0.6 mg/kg and 1.2 mg/kg doses) received a higher cumulative dose compared to any patient enrolled at doses of ≥ 4.0 mg/kg over the course of 1.4 years and 2.6 years, respectively. At the time treatment was discontinued for these patients, one patient had developed rash and periorbital edema and the other had developed rash and skin hyperpigmentation. While patient numbers are limited, the higher cumulative doses tolerated by these two patients compared to the patients at doses of ≥ 4.0 mg/kg suggests that safety events are not only related to cumulative exposure but likely also related to the maximum dose administered in a single infusion.

Despite promising anti-tumor activity, the severity, persistence, and non-resolvable nature of the toxicities compelled the decision to discontinue this phase 1 trial. Due to the limited responses at doses <4.0 mg/kg (1 patient at 1.2 mg/kg), it was deemed unlikely to identify a dose with sufficient efficacy and safety to warrant additional development in the clinic. Although ocular, dermatologic and pulmonary findings in the cynomolgus monkey safety study were categorized as minimal, these clinical data suggest that humans are significantly more sensitive to normal tissue toxicities of DHES0815A.

**Table 1 | Adverse events occurring in ≥ 20% of patients regardless of attribution**

| | DHES0815A 0.6 mg/kg (n = 3) | | DHES0815A 1.2 mg/kg (n = 3) | | DHES0815A 2.4 mg/kg (n = 3) | | DHES0815A 4.0 mg/kg (n = 3) | | DHES0815A 6.0 mg/kg (n = 2) | | All Patients (N = 14) | |
|---|---|---|---|---|---|---|---|---|---|---|---|---|
| | Gr 1-2 | Gr 3-4 | Gr 1-2 | Gr 3-4 | Gr 1-2 | Gr 3-4 | Gr 1-2 | Gr 3-4 | Gr 1-2 | Gr 3-4 | Gr 1-2 | Gr 3-4 |
| Fatigue | | | 1 (33%) | | 2 (67%) | | 1 (33%) | | 1 (50%) | 1 (50%) | 5 (36%) | 1 (7%) |
| Pruritis | | | 1 (33%) | | | | 3 (100%) | | 1 (50%) | | 5 (36%) | |
| Rash | 1 (33%) | | 1 (33%) | | | | 2 (67%) | | 1 (50%) | | 5 (36%) | |
| Blood alkaline phosphatase increased | 1 (33%) | | 2 (67%) | | | | | | 1 (50%) | | 4 (29%) | |
| Dyspnea | 1 (33%) | 1 (33%) | 1 (33%) | | | | | | 1 (50%) | | 3 (21%) | 1 (7%) |
| Headache | 1 (33%) | | | | | | 2 (67%) | | | 1 (50%) | 3 (21%) | 1 (7%) |
| Nausea | | | | | 2 (67%) | | 1 (33%) | | 1 (50%) | | 4 (29%) | |
| Arthralgia | 1 (33%) | | 1 (33%) | | 1 (33%) | | | | | | 3 (21%) | |
| Constipation | 1 (33%) | | | | 1 (33%) | | 1 (33%) | | | | 3 (21%) | |
| Decreased appetite | 1 (33%) | | | | 1 (33%) | | 1 (33%) | | | | 3 (21%) | |
| Photophobia | | | | | | | 1 (33%) | 1 (33%) | 1 (50%) | | 2 (14%) | 1 (7%) |
| Skin hyperpigmentation | | | 1 (33%) | | | | 2 (67%) | | | | 3 (21%) | |
| Urinary tract infection | 1 (33%) | | 1 (33%) | | | | 1 (33%) | | | | 3 (21%) | |
| Pain | | | 1 (33%) | | 1 (33%) | | 1 (33%) | | | | 3 (21%) | |

Adverse events related to study drug by highest NCI CTCAE grade; safety-evaluable patients.

## Discussion

Antibody-mediated delivery of potent cytotoxic agents selectively to target-expressing tumor cells while sparing normal tissue is the rationale for developing ADCs for cancer therapy. With the increasing number of ADC approvals over the last several years[35], the potential for ADCs in clinical use has been realized. However, despite the goal of tumor-selective delivery of cytotoxic agents, ADC-related toxicities remain a significant challenge. Most, but not all, toxicities are off-target —i.e., toxicities in tissues that express little or no antigen—and are due mostly to target-independent uptake into normal tissue. The spectrum of toxicities is ultimately related more to cytotoxic payload than to target expression. Toxicities with ADCs comprised of the tubulin binding agent MMAE (monomethyl auristatin E) across different tumor targets are typically neutropenia and peripheral neuropathy, while ocular toxicities are frequent with ADCs using MMAF (monomethyl auristatin F) or DM4 as payloads[35]. In contrast, the most common T-DM1 related AEs are transaminitis and thrombocytopenia[12]. TEAEs with T-DXd, an ADC with the same antibody as T-DM1 but with a topoisomerase 1 inhibitor payload, include alopecia, cytopenias and interstitial lung disease[14]. As with chemotherapeutic agents, managing ADC toxicities is key to patient care and quality of life.

For designing a next generation HER2 ADC, we selected an antibody that does not compete with trastuzumab or pertuzumab, a stable hindered disulfide linker and a DNA damaging agent as the payload. Early exploration of PBD dimer payloads for this ADC indicated that high potency of the PBD dimer would limit dosing, thereby limiting the TI, and would not allow exposure to achieve maximal efficacy. A number of PBD dimer ADCs have undergone preclinical and clinical evaluation, yet only one is currently approved - loncastuximab tesirine, targeting CD19 for non-Hodgkin Lymphoma. The low doses used, 150 µg/kg (Q3W X 2), followed by maintenance dosing with 75 µg/kg, reflect the high potency of the PBD dimer payload. Thus, we made different chemical modifications to convert the bis-alkylating PBD dimer SG2057 to a mono-alkylating PBD dimer, allowing DNA alkylation with no inter-strand cross-linking. This approach was previously explored with a similar bis-alkylating agent, indolinobenzodiazepine dimer (IGN) for a number of different targets[36,37]. Results with the monoimine IGN were similar to ours with the PBD monoamine in that the ADCs retained potency similar to the parent dimers. The advantages with the IGN monoalkylators were two-fold: enhanced bystander activity and improved therapeutic index. As our desired outcome is a reduction in potency compared to the PBD dimer, we did not perform

safety studies with the monoamine (or other modified PBDs described). More recently, preclinical studies on reduced potency DNA bis-alkylating PBD dimer ADCs were reported[38]. Similar to our findings, higher doses of these agents were required for in vivo efficacy compared to the parent PBD dimer, and the reduced potency ADCs appeared to be tolerated. However, the safety studies described were single dose only. Due to the nature of delayed toxicities with this class of payloads, single dose safety studies are not sufficient to determine tolerability. The PBD monoamide satisfies the criteria set forth— reduced potency, DNA monoalkylation, active as an ADC (DHES0815A) in the desired dose range for optimal PK and efficacy, efficacy in T-DM1 refractory models and in HER2-low mouse tumor models, and an acceptable safety profile when tested in non-human primate safety studies.

In the phase 1 dose-escalation study, early signs of DHES0815A clinical activity are evident. Reaching a dose of 6 mg/kg is in line with the dose level we hope to achieve. However, as treatment of patients continued, a number of concerning toxicities emerged. The AE profile for DHES0815A shares similarities to SYD985, a trastuzumab-based ADC with a DNA mono-alkylating payload, duocarmycin[39,40]. In contrast to DHES0815A, duocarmycin is significantly more potent than PBD-monoamide, thus resulting in a lower clinical dose (1.2 mg/kg Q3W) and modest clinical response (ORR, objective response rate, of 33%). In a similar patient population, the ORR for trastuzumab deruxtecan is 61%, likely reflecting the lower potency of the payload DXd allowing for high DAR and high clinical dosing (5.4 mg/kg Q3W[14],).

Although most DHES0815A-related AEs are categorized as grade 1-2, the severity, persistence, and non-resolvable nature of the toxicities observed at doses of 4.0 mg/kg and higher compelled us to reduce the dose of patients still on treatment to 2.4 mg/kg, discontinue enrollment, and ultimately close the trial. Our hypothesis regarding the observed toxicities is that, although PBD-monoamide does not cross-link DNA, monoalkylation can result in DNA damage, most notably after repeat dosing. At the higher doses of DHES0815A, DNA damage likely accumulates in certain tissues, resulting in toxicity signals. However, it is surprising that the nature of these toxicities in patients was marked, compared to cynomolgus monkeys, given that we allowed for an extended recovery period in the cynomolgus monkey safety study.

In the landscape of ADC development, there are certainly more failures than successes. However, reasons for discontinuation of preclinical or clinical evaluation of ADCs are rarely made public or

published. We believe that the learnings herein hold value to the ADC and oncology communities at large. To design more efficacious and better tolerated ADCs, sharing information regarding discontinuations, challenges, as well as successes will benefit investigators in this field.

## Methods

All mouse xenograft studies were approved by Genentech and Institutional Animal Care and Use Committee (IACUC) and adhered to the NIH Guidelines for the Care and Use of Laboratory Animals. Studies in cynomolgus monkey were performed in accordance with the U.S. Department of Health and Human Services, Food and Drug Administration, United States Code of Federal Regulations, Title 21, Part 58: Good Laboratory Practice for Nonclinical Laboratory Studies in accordance with OECD Principles of Good Laboratory Practice.

### Cell lines and reagents

The following cell lines were from the American Type Culture Collection (ATCC): SK-BR-3 (HTB-30), MCF-7 (HTB-22), HCC1954 (CRL-2338), HCC1569 (CRL-2330), MDA-MB-468 (HTB-132), BT-474 (HTB-20). WSU-DLCL2 (ACC-575) cells were from the German Collection of Microorganisms and Cell Culture (DSMZ). MCF7/Nuclight red and SK-BR-3/H2B-GFP were made at Genentech, and KPL-4 cells were from Dr. J. Kurebayashi[41]. HCC1569 X2 are HCC1569 breast cancer cells that underwent 2 rounds of in vivo selection[42]. Normal cell lines were obtained from LifeLine Cell Technology LLC and included: adult human keratinocytes (catalog #FC-0025), human mammary epithelial cells (#FC-0065), human renal proximal tubule epithelial cells (#FC-0013), human umbilical vein endothelial cells (#FC-0003), and human small airway epithelial cells (#FC-0016). None of the cell lines used for this manuscript are on the Register of Misidentified Cell Lines (International Cell Line Authentication Committee, ICLAC). All cell lines were routinely tested for Mycoplasma using short tandem repeat (STR) profiling and single nucleotide polymorphism (SNP) fingerprinting and were found negative for Mycoplasma contamination. Cells were maintained in Ham's F-12: high glucose DMEM (50:50) supplemented with 10% heat-inactivated fetal bovine serum and 2 mmol/L L-glutamine (all from Thermo Fisher). Normal human cells were cultured according to vendor instructions.

PBD (pyrrolobenzodiazepine) dimers (PBD/SG2057[24], PBD-monoamine, -monoamide, -morpholine, and -olefin derivatives), DXd[17] and T-DXd were synthesized at Genentech. SN-38 was purchased from Tocris. HER2 antibodies (humanized 7C2/hu7C2, trastuzumab, pertuzumab, anti-CD22) and T-DM1 were made at Genentech. PBD-containing ADCs were conjugated to linker-drugs through engineered cysteines at light chain K149 using a nitropyridyl methyl-disulfide linker[22] at Genentech. Materials made at Genentech are available upon request (https://www.gene.com/scientists/mta).

### DNA alkylation

The parent PBD dimer SG2057 and PBD mono-alkylator derivatives were assessed for in vitro alkylation of double-stranded DNA oligonucleotides as follows: PBD dimer analogues (50 μM) were incubated with 50 μM double strand oligonucleotides for 1 h in 10 mM Bis-Tris, pH 7.1 at 37° C. The model nucleotides were 5′ TATAGAAATCTATA 3′ and 3′ATATCTTTAGATAT 5′ (synthesized at Genentech). PBD dimer and saturated dimer were included as positive and negative controls, respectively. The samples were analyzed by LC/MS on Sciex TripleTOF 5600 on a Hypersil Gold C18 column (100x2.1, 1.9 μM, Thermo Scientific). The column was eluted at 0.4 mL/min by a gradient of buffer A (50 mM hexafluoro-isopropanol and 15 mM diethylamine) to buffer B (50% A and 50% of 1:1 methanol:acetonitrile), 5% to 25% B in 8 min, to 75% B in 5 min, and to 95% B in 1 min[43]. Under these conditions, the starting oligonucleotides were eluted at approximately 17.0-18.5 min

and the compound-oligonucleotide adducts were eluted at approximately 24.0-26.0 min.

### Cell viability, apoptosis, ADCC and cell cycle analysis assays

Cells were seeded into 96-well plates and allowed to adhere overnight. Media were removed and replaced with fresh media containing different concentrations of free drugs or conjugates. After specified time periods, Cell Titer-Glo or Caspase-Glo 3/7 (Promega Corp.) reagents were added and luminescence recorded using an EnVision Multilabel Plate Reader (PerkinElmer).

For screening free drugs, both solid tumor and hematologic cancer cells were treated for 3 days with the PBD dimer, PBD-monoamine, -monoamide, -olefin, or -morpholine. Cell line screens for the PBD dimer were performed 7 times (with $n = 56$, $n = 256$, $n = 50$, $n = 383$, $n = 144$, $n = 73$ and $n = 643$ cell lines in the 7 screens). Three screens were performed for the PBD-monoamide ($n = 124$, $n = 146$ and $n = 73$ cell lines). Two cell line screens were performed for the mono-amine ($n = 147$ and $73$ lines), olefin ($n = 145$ and $n = 71$ lines) and morpholine ($n = 147$ and $n = 72$ lines) derivates. Data for the cell line screens are provided as a Source Data file. All cells for screening were from an internal Genentech cell bank. Cells were maintained in RPMI-1640, 5% fetal bovine serum, and 2 mM L-glutamine in a humidified incubator maintained at 37° C with 5% $CO_2$. Cells were assessed with a Vi-CELL Cell Viability Analyzer (Beckman Coulter; Brea, CA); viability of at least 90% was required for screening. A Multidrop™ Combi Reagent Dispenser (Thermo Scientific; Waltham, MA) was used to plate cells into Falcon® 384-well, black, clear-bottom plates (Catalog No. 353962; Corning; Tewksbury, MA) using seeding densities previously determined to achieve approximately 70-80% confluence at the final time point of the assay. The following day, cells were treated with a 9-point dose titration of the PBD molecules using a Bravo Automated Liquid-Handling Platform (Agilent; Santa Clara, CA). After 3 days, 25 μL Cell-Titer-Glo® reagent was added using a MultiFlo™ Microplate Dispenser (BioTek; Winooski, VT). Luminescence was read by a 2104 EnVision® Multilabel Plate Reader (PerkinElmer; Waltham, MA). The data were processed using Genedata Screener®, Version 14 (Genedata; Basel, Switzerland), with a four-parameter Hill equation using compound dose−response data normalized to the median of 42 vehicle-treated wells on each plate. The reported absolute $IC_{50}$ is the dose at which cross-run estimated inhibition is 50% relative to DMSO control wells.

Experiments for ADC potency were performed on breast cancer cell lines SK-BR-3, KPL-4, BT-474, HCC1569, HCC1954, MDA-MB-468 and MCF7 treated for 5 days. Kinetic apoptosis measurements were performed on IncuCyte Live-Cell Analysis System (Essen Bioscience) with CellEvent Caspase3/7 Green substrate (Thermo Fisher). Caspase-3/7 activation was assessed utilizing the IncuCyte Zoom® live cell imaging software. Caspase activation assays were performed on SK-BR-3 and KPL-4 cells. ADCC assays using SK-BR-3 and KPL-4 cells were performed with purified human PBMCs and tumor cells as previously described[7]. To assess the effect of PBD-monoamide on dividing vs. non-dividing cells, growth arrest was induced in SK-BR-3 breast cancer cells by serum-deprivation in serum-free medium, and in normal human adult keratinocytes by contact inhibition. Viability was assessed 3 days after drug treatment with Cell Titer-Glo. Treatment groups for the above cell assays were in quadruplicate (4 wells per treatment group).

For cell cycle analysis, breast tumor cells were seeded into 20 x 100 mm tissue culture dishes and allowed to adhere overnight. Cells were then exposed to PBD-monoamide or DHES0815A for 48 h at 37° C. Both floating and adherent cells were harvested, fixed with cold methanol, and stained with propidium iodine/RNase staining buffer (BD Biosciences). The stained cells were analyzed by flow cytometry on a FACSCalibur instrument (BD Biosciences) using ModFit LT software (Verity Software House). One culture dish per treatment group was

used to generate cell cycle histograms; cell cycle experiment was performed 3 times with similar results.

## Bystander cell killing assays

MCF7/Nuclight red and SK-BR-3/H2B-GFP (histone 2B-green fluorescent protein) stable cell lines were treated individually or co-incubated 1:3 with different HER2 ADCs (400 ng/mL) in 6-well plates for 5 days ($n$ = 3 per group). Cells were then collected by trypsinization and the total number of live cells determined using ViCell cell counter (Beckman Coulter). The ratio of SK-BR-3/H2B-GFP and MCF7/Nuclight red in each well was measured by FACS (flourescence activated cell sorting) using the Fortessa Flow Cytometer (Becton Dickinson). From the total number of live cells and the ratio of the two cell lines, the number of viable SK-BR-3/H2B or MCF-7/Nuclight red cells was calculated.

## Immunoblot procedures

SK-BR-3 cells were seeded at a density of 1 million per dish in 100 x 15 mm dishes and allowed to adhere for 2 days. The medium was then removed and replaced with fresh medium containing either free drug, trastuzumab, T-DM1, MHES0488A or DHES0815A. Following a 48 h incubation, floating cells were collected and combined with detached adherent cells. Cells were lysed in ice-cold cell extraction buffer (Invitrogen, catalog# FNN0011) supplemented with protease inhibitor cocktail tablet (Roche, catalog#14583920), phosphatase inhibitor cocktail 2 and 3 (Sigma, catalog# P5726 and P0044). Lysates were cleared by centrifugation at 4° C for 10 min at 14,000 rpm in a microcentrifuge, and protein concentrations were determined using the BCA Protein Assay Kit (Pierce, catalog# 23225). Proteins were resolved by SDS-PAGE (NuPAGE 4-12% Bis-Tris Gels, ThermoFisher, catalog# WG1402BOX), transferred to nitrocellulose (ThermoFisher iBlot 2 Dry Blotting System), and immunoblotted with the indicated primary antibodies. Antibodies (1:1000 dilution) included phospho-H2AX (Ser139) (catalog#2577), phospho-p53 (Ser15) (catalog# 9286), p53 (catalog# 2527), phospho-Histone H3 (Ser10) (catalog# 9701), PARP (catalog# 9541) and b-actin (catalog# 5125), all obtained from Cell Signal Technology. Blotting was carried out in Tris-buffered saline containing 0.1% Triton X-100 and 5% nonfat dry milk, followed by incubation with horseradish peroxidase-conjugated secondary antibodies (Amersham Biosciences). Proteins were visualized using enhanced chemiluminescence reagents (Amersham Biosciences). Unprocessed scans of blots can be found in the Source Data file.

## Scatchard analysis

Humanized anti-HER2 7C2 (hu 7C2) was radiolabeled with $^{125}$I for a specific activity of 9.71 μCi/μg for studies on BT-474 cells, which overexpress human HER2. Specific binding was assessed using unlabeled ('cold') hu 7C2, starting at 1000 nM followed by 1:3 serial dilutions. For studies on cynomolgus monkey HER2, CHO (Chinese Hamster Ovary) cells were engineered to express full length cynomolgus monkey HER2. $^{125}$I hu 7C2, specific activity 34 μCi/μg, was added to cells in the presence of cold hu 7C2, at a concentration of 500 nM, with 1:3 serial dilutions. Cells were incubated at room temperature for 2 hr. Radioactivity was measured using a Packard Gamma Counter (Perkin Elmer).

## Fluorescence-activated cell sorting (FACS)

Cells were harvested with cell dissociation buffer, washed with PBS containing 2% FBS and incubated with the appropriate primary and secondary antibodies. Binding of murine 7C2, humanized 7C2 and anti-neu MAb 2009[21] was assessed on DHFRG8 cells (American Type Culture Collection, ATCC) using 10 μg/mL of each antibody and detecting with anti-human or anti-rat/mouse PE secondary antibodies. To measure potential HER2 antibody (trastuzumab, pertuzumab) cross-blocking of MHES0448A binding, SK-BR-3 cells were harvested with cell dissociation buffer, washed with PBS/2% FBS and incubated with

10 μg/mL primary antibodies MHES0448A, pertuzumab or trastuzumab for 1 h at 4° C. Cells were washed twice and labeled with 10 mg/mL Alexa Fluor 488-MHES0448A for 1 h at 4° C. The labeled cells were washed and analyzed by flow cytometry on the Guava EasyCyte (Guava Technologies) using FlowJo software (FlowJo, LLC).

## Fluorescence microscopy

Cells were seeded onto 8-well LabTekII microscope slides (Nalge Nunc) and incubated at 37° C for 19 h with 1 μg/mL DHES0815A, or unconjugated antibodies in growth medium containing lysosomal protease inhibitors (Roche) to minimize antibody degradation. Cells were then washed, fixed in 3% paraformaldehyde, quenched with 50 mM NH$_4$Cl, and permeabilized with saponin buffer. The endosomal and lysosomal compartments were visualized by staining with mouse anti-human LAMP1 (lysosomal-associated membrane protein 1). HER2 and LAMP1 antibodies were detected with Cy3 anti-human and Alexa647 anti-mouse, respectively. Cells were imaged by deconvolution microscopy using a 60X Plan Apo N oil objective (Olympus) of numerical aperture 1.42 and a DAPI/FITC/TRITC/Cy5 filter set on a DeltaVision microscope (Applied Precision) controlled by Resolve3D software (SoftWoRx). Images were captured with a CoolSNAP_HQ2 CCD camera (HQ2-ICX285) and single slices were deconvolved for 10 cycles (aggressive). Figures were compiled using Adobe® Photoshop® CS5, version 12.1 x 64 (Adobe Systems).

## Mouse xenograft studies

MMTV-HER2 Fo5, HCC1954 X2 and WHIM8 xenograft studies were performed at Genentech. HCC1954 X2 cells (5 million) in 50% matrigel/HBSS were implanted into the #2/3 mammary fat pads of female C.B-17 SCID.bg mice from Charles River Laboratories (11 weeks of age for studies in Figs. 4a and 4d T-DM1 combination; 16 weeks for Fig. 4d docetaxel combination). Tumor tissue from MMTV-HER2 Founder 5 (Fo5) HER2 transgenic mice[11] was collected aseptically, rinsed in HBSS, cut into pieces of approximately 2x2 mm in size and cryopreserved for use in later studies. Fragments were surgically transplanted into the mammary fat pad of female CRL Nu/Nu mice (Charles River Laboratories, Hollister, CA), 11 weeks of age for studies in Figs. 1d and 4d (T-DM1 combination) or 13 weeks for Fig. 4d (docetaxel combination). WHIM8 PDX (patient-derived xenograft) tumors[44] were dissociated in RPMI medium containing insulin (10 ng/mL), EGF (10 ng/mL), hydrocortisone (10 μg/mL), collagenase (0.5 mg/mL), hyaluronidase (0.1 mg/mL) and 5% FBS prior to implantation in mammary fat pads of female NSG (NOD SCID gamma) mice (The Jackson Laboratory, 1 million cells in 50% matrigel/HBSS). Mice were 13 weeks of age for the study in Fig. 4a and 11 weeks old for the study in Fig. 5b.

Gastric (STO410 and STO41) and breast (BC207, BC197, BC128 and BC085) cancer PDX models were carried out at GenenDesign (Shanghai, China). Mice were 6 weeks of age for the gastric and breast cancer PDX models. Tumor fragments were implanted subcutaneously in the right flanks of Balb/c nude mice (Vital River) and allowed to reach 150-300 mm³ before starting treatment.

WSU-DLCL2 B-cell lymphoma cells were inoculated subcutaneously into 12 week old C.B-17 Fox Chase SCID mice (20 million cells per mouse in HBSS). When tumors reached 100-300 mm3, mice were randomized into 7 groups of 5 mice each. Vehicle buffer or different ADCs were administered as a single intravenous injection and tumors measured 1-2 times per week.

For all studies, tumor and body weight measurements were taken 2 times per week until end of study. A Linear Mixed Effects (LME) model, using a package of customized functions in R version 3.6.2 (R Foundation for Statistical Computing), was used to generate fitted curves for tumor volume changes over time[45]. This model allows analysis of natural log transformed tumor volume changes over time and addresses both repeat measurements from the same animal and accounts for dropouts before the end of the study. Maximum

allowable tumor volume was 2000 mm³ for all studies. Tumor growth curves (Fig. 4 b and c) depicting tumor volume > 2000 mm³ are the result of the program generating a best fit curve at all time points.

Mice were housed in individually ventilated cages within animal rooms maintained on a 14:10-hour light:dark cycle. Animal rooms were temperature and humidity-controlled, between 68 – 79 °F (20.0 – 26.1° C) and 30 – 70% respectively, with 10 to 15 room air exchanges per hour.

### HER2 Immunohistochemistry (IHC) on xenograft tumors

IHC was performed on 4-μm thick formalin-fixed, paraffin-embedded xenograft or patient-derived xenograft sample sections mounted on glass slides. For HER2, staining was performed on the Ventana Discovery XT Autostainer platform (Ventana Medical Systems Inc). The slides were pretreated with CC1 antigen retrieval buffer, standard time, followed by anti-HER2 (clone D8F12 Cell Signaling Technology #4290), incubated at 0.1 μg/mL for 16 min at 37° C. The antibody was detected with anti-rabbit-UltraMap (Ventana Medical Systems Inc). Staining was visualized with DAB (diaminobenzidine). Sections were counterstained with hematoxylin, dehydrated, cleared, and coverslipped for viewing.

### Safety studies in cynomolgus monkey.

Safety studies in cynomolgus monkey (*Macaca fascicularis*) were performed at Charles River Laboratories (Reno, NV). DHES0815A was administered IV at doses of 0 (vehicle), 4, 8 and 12 mg/kg Q3W for 5 treatments (3 females and 3 males per group), followed by a 7-week recovery period (2 females and 2 males per group). Vehicle was 20 mM histidine acetate, 240 mM sucrose, 0.02% (w/v) polysorbate 20, pH 5.5. Animals were monitored for clinical observations (moribundity, body weight, food intake, ophthalmic examinations). Electrocardiology, cardiovascular parameters via telemetry, neurologic examinations, clinical pathology, hematology and clinical chemistry were also included. Blood plasma was collected for toxicokinetic analysis pre- and post-dose (0.25 and 24 hr) on day 1; post-dose on days 8 and 15; and pre- and 0.25 hr post-dose on days 22 and 43; pre- and post-dose (0.25 and 24 hr) on day 64; post-dose on days 71 and 78; pre- and post-dose (0.25 hr) on day 85; and post-dose on days 92, 106, 120 and 134.

### Pharmacokinetic (PK) assays in cynomolgus monkey.

DHES0815A conjugate was measured as antibody-conjugated PBD-monoamide (acPBD-ma) using a method consisting of immunoaffinity (IA) capture of DHES0815A from plasma with magnetic Protein A beads, followed by chemical reduction/alkylation to release PBD-monoamide from DHES0815A, and LC-MS/MS for detection. A validated method was used for the GLP cynomolgus monkey toxicokinetic (TK) analysis. DHES0815A total antibody (all DAR species, including DAR 0 and DAR ≥ 1) was measured by a validated IA LC-MS/MS method for the GLP cynomolgus monkey TK analysis and unconjugated PBD-monoamide was measured using LC-MS/MS.

### Phase 1 study

#### Study Oversight.

This study was a first-in-human, phase I, open-label, multicenter, traditional 3 + 3 dose-escalation study conducted in accordance with the Declaration of Helsinki, the principles of Good Clinical Practice, and in compliance with all relevant ethical regulations. The protocol was approved by institutional review boards and ethic committees from each study site (Dana Farber Cancer Institute, Memorial Sloan Kettering Cancer Center, Yale Cancer Center, Asan Medical Center, Columbia Medical Center, and Sarah Cannon Research Institute), and all patients provided written informed consent before undergoing any study procedures. Patient provided consent to publish the images in Supplementary Fig. 16. Patients were not compensated for participation in the study. This trial is registered at ClinicalTrials.gov with trial number: NCT03451162. Study enrollment dates were 17 April 2018 to 11 April 2019.

### Study design and endpoints.

The primary objective was to evaluate safety and tolerability of escalating doses of DHES0815A administered by IV infusion Q3W to patients with advanced or metastatic HER2+ breast cancer, including estimation of the maximum tolerated dose (MTD), determination of the recommended phase II dose (RP2D), and characterization of dose-limiting toxicities (DLTs) assessed according to the Common Terminology Criteria for Adverse Events (CTCAE) v4.0 (NCI CTCAE v4.0, 2010). Secondary objectives included evaluating pharmacokinetics, objective response rate (ORR), duration of response (DOR), and incidence of anti-DHES0815A antibodies.

Eligible patients were age 18 years or older with Eastern Cooperative Oncology Group (ECOG) performance status of 0 or 1 with at least one measurable target lesion per RECIST v1.1. Patients were required to have locally advanced or metastatic HER2+ breast cancer (meeting ASCO/CAP 2013 definition) that had relapsed or was refractory to established therapies, and adequate hematologic and end-organ function.

The study utilized a 3 + 3 dose-escalation design to define the MTD/maximum administered dose (MAD) of DHES0815A administered by IV infusion on Day 1 of a 21-day cycle. Patients were closely monitored for adverse events during the DLT window, defined as Days 1-21 of Cycle 1. The starting dose of DHES0815A was 0.6 mg/kg. Intrapatient dose escalation was permitted to minimize exposure at suboptimal doses and to maximize the collection of information at relevant doses, provided that patients completed at least three cycles at their original assigned dose level as well as at each subsequent dose level, prior to any further dose escalation.

### Data collection.

Study visits occurred between April 2018 and July 2021 and data were entered into electronic case report forms (eCRF). Specific sites: Columbia University Medical Center, New York, NY; Memorial Sloan Kettering Cancer Center, New York, NY; Asan Medical Center, University of Ulsan College of Medicine, Seoul, Korea; Sarah Cannon Research Institute/Tennessee Oncology, Nashville, TN; Yale Cancer Center, Yale University, New Haven, CT.

### Assessments.

Response evaluations were assessed using RECIST v1.1 at baseline and approximately every 6 weeks until cycle 8, then every 9 weeks thereafter until treatment discontinuation, and at the end of treatment visit. Response was assessed using image-based evaluations per CT (computerized tomography) or MRI (magnetic resonance imaging). Samples for pharmacokinetic analysis were collected to capture DHES0815A exposure data at a sufficient number of time-points to provide a detailed profile of the concentration-time curve for DHES0815A total antibody, antibody-conjugated PDB-monoamide (acPDB-MA), and unconjugated PBD-monoamide. Safety was monitored continuously throughout the study period and during the 42-day safety follow-up period after end of treatment visits.

HER2-targeting molecules increase patient risk of developing left ventricular dysfunction. Left ventricular function was regularly monitored by echocardiogram or MUGA (multi-gated acquisition) scans in all patients treated with DHES0815A.

Because ocular toxicities and corneal pigmentation were observed after administration of DHES0815A to cynomolgus monkey, ocular toxicities were classified as a potential risk where patients may present with impaired vision. Therefore, ophthalmic examinations were required during the screening period, every 6 weeks for the first 6 cycles and every 12 weeks thereafter, and at the study discontinuation visit.

Interstitial lung disease has been reported in patients receiving HER2-targeted agents. Therefore, patients with clinically significant pulmonary symptoms or diseases were therefore excluded from this study. Chest CT scans were reviewed in patients who presented with signs or symptoms indicative of changes in pulmonary function.

The safety and efficacy analysis population included all subjects who received at least 1 cycle of therapy (1 dose of DHES0815A Q3W).

The trial intent was to enroll HER2+ metastatic breast cancer patients agnostic of gender. Given the small number of patients, early termination and rarity of male breast cancer, the enrolled patients were all female. No sex or gender analyses were carried out.

**Pharmacokinetic analyses.** The pharmacokinetics of DHES0815A were characterized by measuring DHES0815A total antibody (all DARs including fully conjugated, partially deconjugated, and fully deconjugated anti-HER2 antibody) and antibody-conjugated PBD-monoamide by immunoaffinity LC-MS/MS, and unconjugated PBD-monoamide by LC-MS/MS[46]. Total exposure (area under the concentration-time curve), maximum observed serum concentration, minimum observed serum concentration, clearance, and volume of distribution at steady state were derived pharmacokinetic parameters.

**Statistics and reproducibility.** For cell-based experiments, all studies were repeated between 2-6 times with similar results. Data presented are pooled data from multiple studies. For mouse xenograft studies, most studies were performed once, with the exception of DHES0815A dose-response studies (Fig. 4a), which were repeated 2-3 times.

For the clinical study, no statistical method was used to pre-determine sample size. The exact number of patients to be enrolled in the study was to depend upon the observed safety and pharmacokinetic/pharmacodynamic profile according to the 3 + 3 dose escalation design and dose escalation rules. No data were excluded from the analyses. The experiments were not randomized. The Investigators were not blinded to allocation during experiments and outcome assessment.

### Reporting summary

Further information on research design is available in the Nature Portfolio Reporting Summary linked to this article.

## Data availability

To request access to clinical data that support the findings of this study, see here (https://vivli.org/members/enquiries-about-studies-not-listed-on-the-vivli-platform/), and for up-to-date details on Roche's Global Policy on the Sharing of Clinical Information, see here (https://www.roche.com/innovation/process/clinical-trials/data-sharing/). Roche provides qualified researchers access to individual patient data; data are available for up to 24 months once approved. Remaining data can be found in the Article, Supplementary, and Source Data files. The Study Protocol is provided in the Supplementary Information file as Supplementary Note 1. Source data are provided with this paper.

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

## Acknowledgements

Study design, data collection and analysis, and manuscript writing were sponsored by Genentech. We would like to acknowledge Bryan Hains for assistance with manuscript preparation, and Shari Lau for immunohistochemistry support. In addition, we thank all of the study participants and their families, and all of the site investigators, study coordinators, and staff.

## Author contributions

Study concept and design: G.D.L., G.L., J.G., S.-F.Y., C.T.F., D.Z., P.S.D., T.P., A.U., L.C., M.C.S., E.C. Acquisition, analysis, or interpretation of data: G.D.L., G.L., J.G., S.-F.Y., C.T.F., D.Z., D.L., M.V.L., O.S., V.C., A.U., S.M., L.C., M.M.S., R.C., M.X.S., E.C. Statistical analysis: G.L., C.T.F., S.-F.Y., D.L., V.C. Study supervision: G.D.L., S.-F.Y., D.L., A.U., L.C., K.K., S.M., K.H.J., E.H., P.L., I.K., M.M.S., R.C., E.C. Drafting of the manuscript: All authors. Final approval of manuscript: All authors

## Competing interests

G.D.L., G.L., J.G., S.-F.Y., C.T.F., G.L., D.Z., P.S.D., T.P., B.W., D.L., A.K., T.W., M.V.L., O.S., V.C., A.U., S.M., L.C. and E.C. are full-time employees of Genentech and shareholders in F. Hoffman-La Roche, Ltd. Jack Sadowsky is a former employee of Genentech, a shareholder in F. Hoffmann-La Roche, Ltd., and an employee of Carmot Therapeutics. Michael Mamounas is a former employee of Genentech and shareholder in F. Hoffmann-La Roche Ltd. K.K. is an advisor and consultant for Eli Lilly, Pfizer, Novartis, Astra Zeneca, Daiichi Sankyo, Puma, 4D Pharma, Oncosec, Immunomedics, Merck, Seagen, Mersana, Menarini Silicon Biosystems, Myovant and Takeda; and receives institutional research funding from Genentech/Roche, Novartis, Eli Lilly, AstraZeneca, Daiichi Sankyo, and Ascentage. S.M. is an advisor and consultant for Genentech, AstraZeneca, Daiichi Sankyo, Macrogenics, Gilead and Seagen; and receives institutional research funding from Genentech, Astra Zeneca, Daiichi Sankyo and Seagen. K.H.J. is an advisor and consultant for AstraZeneca, Bixink, Novartis, Roche, MSD, Pfizer, Everest Medicine, Daiichi Sankyo, Eisai, and Takeda. E.H. is an advisor and consultant (all payments to institution) for Arcus, AstraZeneca, Daiichi Sankyo, Deciphera Pharmaceuticals, Ellipses Pharma, Genentech/Roche, Greenwich LifeSciences, iTeos, Janssen, Eli Lilly, Loxo, Mersana, Novartis, Olema Pharmaceuticals, Orum Therapeutics, Pfizer, Relay Therapeutics, Seagen, Stemline Therapeutics, Verascity Science; and receives institutional research funding from, Abbvie, Acerta Pharma, Accutar Biotechnology, ADC Therapeutics, AKESOBIO Australia, Amgen, Aravive, Artios, Arvinas, AstraZeneca, AtlasMedx, BeiGene, Black Diamond, Bliss BioPharmaceuticals, Boehringer Ingelheim, Cascadian Therapeutics, Clovis, Compugen, Cullinan-Florentine, Curis, CytomX, Daiichi Sankyo, Dana Farber Cancer Inst, Dantari, Deciphera, Duality Biologics, eFFECTOR Therapeutics, Ellipses Pharma, Elucida Oncology, EMD Serono, FujiFilm, G1 Therapeutics, Genentech/Roche, H3 Biomedicine, Harpoon, Hutchinson MediPharma, Immunogen, Immunomedics, Incyte, Infinity Pharmaceuticals, InventisBio, Jacobio, Karyopharm, K-Group Beta, Eli Lilly, Loxo Oncology, Lycera, Mabspace Biosciences, Macrogenics, MedImmune, Mersana, Merus, Millennium, Molecular Templates, Novartis, Nucana, Olema, OncoMed, Onconova Therapeutics, Oncothyreon, ORIC Pharmaceuticals, Orinove, Pfizer, PharmaMar, Pieris Pharmaceuticals, Pionyr Immunotherapeutics, Plexxikon, Radius Health, Regeneron, Relay Therapeutics, Repertoire Immune Medicine, Rgenix, SeaGen, Sermonix Pharmaceuticals, Shattuck Labs, StemCentRx, Sutro, Syndax, Syros, Taiho, TapImmune, Tesaro, Tolmar, Torque Therapeutics, Treadwell Therapeutics, Verastem, Vincerx Pharma, Zenith Epigenetics, Zymeworks. P.L.R. currently is on Advisory Boards for I-Mab, Mersana Therapeutics, BAKX Therapeutics, Scenic Biotech, Qualigen, NeuroTrials; and a consultant for Roivant Sciences. I.K. is on the advisory boards and a consultant

for Daiichi Sankyo, Macrogenics, Genentech/Roche, Seagen, AstraZeneca, Novartis and Merck; and receives institutional research funding and grants from Genentech/Roche, Pfizer and Macrogenics. M.S. is a former employee of Genentech and shareholder in F. Hoffmann-La Roche, Ltd. and a current employee and shareholder at Seagen. R.C. is a former employee of Genentech and shareholder in F. Hoffmann-La Roche, Ltd. and a current employee and shareholder at Gilead Sciences, Inc. M.S. is a former employee of Genentech and shareholder in F. Hoffmann-La Roche, Ltd.; is currently a consultant for Genentech; and is on the Scientific Advisory Boards for Novartis Institutes for BioMedical Research (NIBR), Olema Pharmaceuticals, ABL Bio.

## Additional information

**Peer review information** : *Nature Communications* thanks Xichun Hu, Timothy Lowinger and the other, anonymous, reviewer(s) for their contribution to the peer review of this work. A peer review file is available.

Gail D. Lewis[1] ✉, Guangmin Li[1], Jun Guo[1], Shang-Fan Yu[2], Carter T. Fields[3], Genee Lee[2], Donglu Zhang[4], Peter S. Dragovich[5], Thomas Pillow[5], BinQing Wei[6], Jack Sadowsky[7,21], Douglas Leipold[8], Tim Wilson[9], Amrita Kamath[8], Michael Mamounas[10], M. Violet Lee[11], Ola Saad[11], Voleak Choeurng[12], Alexander Ungewickell[13], Sharareh Monemi[13], Lisa Crocker[2], Kevin Kalinsky[14], Shanu Modi[15], Kyung Hae Jung[16], Erika Hamilton[17], Patricia LoRusso[18], Ian Krop[18], Melissa M. Schutten[19,22], Renee Commerford[13,23], Mark X. Sliwkowski[20] & Eunpi Cho[13]

[1]Discovery Oncology, Genentech, South San Francisco, CA, USA. [2]Translational Oncology, Genentech, South San Francisco, CA, USA. [3]US Medical Affairs, Genentech, South San Francisco, CA, USA. [4]DMPK, Genentech, South San Francisco, CA, USA. [5]Discovery Chemistry, Genentech, South San Francisco, CA, USA. [6]Computational Chemistry, Genentech, South San Francisco, CA, USA. [7]Protein Chemistry, Genentech, South San Francisco, CA, USA. [8]Preclinical and Translational Pharmacokinetics, Genentech, South San Francisco, CA, USA. [9]Oncology Biomarker Development, Genentech, South San Francisco, CA, USA. [10]Project Team Leadership, Oncology, Genentech, South San Francisco, CA, USA. [11]Bioanalytical Sciences, Genentech, South San Francisco, CA, USA. [12]Data Sciences, Genentech, South San Francisco, CA, USA. [13]Early Clinical Development, Oncology, Genentech, South San Francisco, CA, USA. [14]Winship Cancer Institute at Emory University, Atlanta, GA, USA. [15]Memorial Sloan Kettering Cancer Center, New York, NY, USA. [16]Asan Medical Center, University of Ulsan College of Medicine, Seoul, Korea. [17]Sarah Cannon Research Institute/Tennessee Oncology, Nashville, TN, USA. [18]Yale Cancer Center, Yale University, New Haven, CT, USA. [19]Safety Assessment Pathology, Genentech, South San Francisco, CA, USA. [20]Molecular Oncology, Genentech, South San Francisco, CA, USA. [21]Present address: Carmot Therapeutics, Berkeley, CA, USA. [22]Present address: SeaGen, South San Francisco, CA, USA. [23]Present address: Gilead Sciences, Foster City, CA, USA. ✉e-mail: gdl@gene.com

