## [Peer Review File · Nature Communications]

Reviewers' Comments:

Reviewer #1:

Remarks to the Author:

It was a pleasure to have the opportunity to review and provide commentary on the manuscript "Development of DHES0815A: a Novel HER2-Directed Antibody-Drug Conjugate Comprised of a Reduced Potency Mono-alkylator Linked to a Domain I Binding HER2 Antibody" by Lewis et al. Antibody drug conjugates are of very high and growing interest, and this manuscript represents a rare example where not only is pre-clinical characterization in vitro and in vivo described, but the authors also share information on the non-clinical tolerability assessment in non-human primates (rarely disclosed) as well as clinical data from the dose escalation study. Together, this provides the reader with an opportunity to compare, contrast, and evaluate the translatability of pre-clinical findings to the observations in the clinic. I do believe that readers interested in the ADC space will find this to be very interesting and to provide helpful insights and perspective on the challenges of ADC discovery and development, and that the manuscript would be of interest to readers generally interested in the field of research into new approaches to the treatment of cancer as well as to readers specifically focused on the field of ADCs.

I do have several comments and/or suggestions for the authors to consider, which I outline below:

- In the abstract it states that DHES0815A was efficacious in in vivo efficacy models and was well tolerated in cynomolgus monkey safety studies. What is most relevant is the predicted therapeutic index, ideally based on exposure. Can the authors share what they believe the non-clinical data predicts as the therapeutic index?
- In the introduction it is stated that a goal was to provide a different treatment regimen other than T-DM1. I believe they mean an alternative treatment option – to me, regimen implies dosing frequency, e.g. qwk vs. q3wk dosing. Clearly the goal here is different.
- There is an emphasis on reducing the potency, whereas I think the real challenge is increasing the tolerability (although I acknowledge the two are related). Also, it is stated that one reason to aim for a higher tolerated dose is to avoid TMDD, but I believe that even with targets where there isn't TMDD as with HER2, for solid tumors the antibody dose matters. Work by Wittrup, Thurber (e.g. Journal of Theoretical Biology 314 (2012) 57–68) highlight this. I would expect that the authors would have no problem with a payload that was picomolar potent as long as it was tolerated at much higher doses in order to allow one to safely dose through any TMDD. Perhaps the authors could better explain their effort as a trade-off on potency in favor of tolerability – i.e. they aren't necessarily increasing the therapeutic index, but rather shifting it into a range that allows for a higher antibody dose (that exceeds the TMDD concern).
- It's stated that only 58% alkylation was observed with the PBD-monoamide". Can the authors comment on why 100% alkylation (or at least closer to 100%) isn't achieved?
- In Figure 1a, the ADC units are listed as nM, but the values provided are ug/mL. Ideally the in vitro concentrations for the ADC would be provided in nM to allow for more facile comparison of the impact of antibody delivery
- Several in vivo efficacy models are utilized, and their HER2 IHC status is indicated. It would be helpful if the antigen expression level for the cell-line derived models could be reported (e.g. is it 500K antigens per cell? 50K? Readers are interested in correlating efficacy with antigen expression)
- In Figure 1D, can the authors comment on why the cell viability for the monoalkylator appears to plateau at 40%? Is this expected given the finding that alkylation only proceeded to ca. 58% (see above)?
- It is indicated that the lack of activity in HER2 negative MCF7 cells indicates target dependence. This would be more convincing if the IC50 of the payload alone in these cells was provided, to confirm that these cells are not insensitive / less sensitive to the payload.
- The statement "HER2+ breast cancer cells showed differential responses to DHES0815A vs. T-DM1 which correlates with intrinsic sensitivity to the two different payloads...". Is there any insight into what causes these intrinsic sensitivities? For the monoalkylator a range of close to 1000x is seen between the most sensitive and the least sensitive cell lines – how does this correlate with antigen expression level, Pgp status, or other possible mechanisms of reduced sensitivity / resistance? Is this concerning for the clinical application?
- The statement "In recent years, it has been appreciated that bystander activity of ADCs is a

desirable attribute" – I would propose that this be softened to "may be a desirable attribute". It is likely highly target and platform dependent.

- In Figure 3, the Figure and caption don't indicate the cells used for the experiment. It would be helpful to indicate this clearly in the Figure/legend. This is particularly important given the wide range of sensitivity of different cells as indicated in Figure 2.
- For the in vivo efficacy studies, it would be very informative to include IHC images that would provide a perspective on the homogeneity as well as level of HER2 expression
- In Figure 4 there are (presumed) errors in the dosing levels – 10 mg/kg is indicated for two lines, whereas no 15 mg/kg line is indicated. Also, Docetaxel is misspelled in 4D.
- The findings of delayed toxicity are very interesting. However, it is unclear whether what is observed is the result of delayed toxicity or cumulative toxicity. Given the (presumed) irreversible DNA alkylation, do the authors have a view on whether the toxicity is truly delayed (i.e. slow to manifest after a single dose) or rather is the result of cumulative toxicity? Does the tolerability of single, high doses vs. lower multiple doses provide any insight to this? What was the recovery time in the Q3wx2 vs. Q3wx4 studies?
- The authors are to be applauded for including NHP tolerability data, as this is rarely shared in publications. Given that only one animal showed focal alveolar degeneration, I would suggest that the statement that these findings were reversible as they weren't observed in recovery animals is perhaps an overstatement; it may be that this is a less frequent finding and is not reversible. Any comments on whether certain toxicities are considered target related or not would also be welcomed, and it would be ideal if a therapeutic index (range) based on AUC for NHP tolerability vs. AUC mouse efficacy would be given.
- In the summary of clinical responses for the 14 patients, data is only described for 13 patients (one CR, 10 SD and 2 PD). What about the 14th patient? Also, in the waterfall plot (Fig 6A) there are more patients that achieved >30% reduction in tumor volume, however they don't qualify as PRs by RECIST. Is this because the tumor reduction was not confirmed on the following scan, or for other reasons? A sentence to explain this would be helpful. Also, the HER2 IHC status is indicated as 2+ or 3+, but there is no indication of the FISH status. It would be helpful to indicate whether or not these patients are considered HER2-positive as opposed to just indicating the IHC score, as IHC-2+ is ambiguous with regard to HER2-positivity by current diagnostic standards.

Overall, I do find this to be a very interesting manuscript that I believe will be of high interest, and I would recommend publication once the manuscript has been revised to correct errors and also to consider the suggestions above. The methodology is sound, and the reporting of both NHP tolerability information as well as patient phase 1 dose escalation information is especially helpful in allowing the reader to evaluate the translatability of non-clinical to clinical results.

I do believe that more detail is required in the methods; for example, at a minimum the specific cell lines used for in vitro experiments need to be disclosed. As noted above, for some experiments (such as the caspase activation) it is not clear which cell line was used.

Signed,
Timothy B. Lowinger, PhD
Chief Science and Technology Officer, Mersana Therapeutics

Reviewer #2:

Remarks to the Author:

This is a good research assessing a novel HER2 ADC DHES0815A. Given the current treatment landscape for mBC is evolving, especially for the approval of T-DXd to treat patients with HER2-low metastatic breast cancer, it is extremely necessary to develop new therapeutic drugs; the manuscript by Lewis et al. is thus of interest. However, the novelty of this article is a little lacking.

1. As an ADC that targets HER2-expression is an already well-established therapeutic strategy in widespread use, and this study of DHES0815A does not clearly demonstrate any evidence of clear superiority of this agent to others available (efficacy & toxicity), we assume this study will not be of great interest to the NC Readership.
2. For HER2 ADCs, it is now known that lower toxic Payload and higher DAR (drug-to-antibody

ratio) may have more chances to be successful. Highly toxic Payload is often difficult to achieve the expected efficacy, and will also produce more risks of delayed toxicity. Although this article contains a lot of information, such as design of DHES0815A, mechanisms of action, efficiency in HER2+and HER2-low xenograft models, safety studies in non-human principles, and phase 1 safety and efficiency data, data volume is large but the results are not ideal.

3. The author did not fully discuss the reasons for the failure of this research and the implications for the future. As authors mentioned, late-onset adverse events, especially those related to the accumulation of dosing, led to close DHES0815A trial, but DS8201 with interstitial lung disease is a success story. A failure of a novel agent may be dependent on other factors, including the modification of anti-HER2 antibody and conjugation method, linker technology (disulfide linker more likely to lead to fall-off of payload in non-targeted tissue), besides the payload cause. Please discuss more on reasons for DHES0815A failure, details of the late-onset AE reported in this manuscript, for example, dyspnea: any CT finding, and presence or absence of ILD? Headache: what's the cause? Any imaging finding? Photophobia: any ocular test results? More important, information on medical interventions and clinical outcomes for those AEs are much needed.

4. We agree that the recent phase I trial design is impossible to define the late-onset fatal AE in the first 28 days after drug administration, but it can be further characterized in the following phase II and III trial

Reviewer #3:

Remarks to the Author:

Detailed, accurate and well-written paper on the development of a novel ADC, namely DHES0815A. Extensive preclinical work. This drug shows toxicities that may be typical to ADCs. However, both preclinical and clinical development (no DLTs in the study) failed to capture a delayed and irreversible toxicity. A mechanistic description/hypothesis explaining why such toxicity occurred later on study and why it was missed, or at least underestimated, in previous phases of development would strengthen the paper, making it more scientifically attractive.

Responses to reviewer's comments, NCOMMS-22-49847-T:

Reviewer #1:

1. In the abstract it states that DHES0815A was efficacious in in vivo efficacy models and was well tolerated in cynomolgus monkey safety studies. What is most relevant is the predicted therapeutic index, ideally based on exposure. Can the authors share what they believe the non-clinical data predicts as the therapeutic index?

Response: *Thank you for pointing out that we did not discuss the therapeutic index. We are cautious regarding discussions around therapeutic index for a number of reasons. The parameters used for calculating TI differ amongst investigators, making it difficult to compare TIs across different molecules. For example, MED (minimum efficacious dose) has been described as a dose that causes any degree of tumor growth inhibition/delay. This will vary with tumor model, and is also dependent on the number of doses administered. Similarly, MTD (or HNSTD) is highly dependent on dosing regimen/frequency, and will be higher with a regimen using fewer doses, which of course impacts TI calculation. Most reported TIs are also not corrected for exposure or body surface area, but are dose-based comparisons of mg/kg MED mouse dose to mg/kg MTD non-human primate dose. A recent paper in Cancer Cell (The Therapeutic Window of Antibody Drug Conjugates: a Dogma in Need of Revision, vol. 40, 2022) highlights the limitations of using TI for ADCs. For our internal purposes, we define MED as the ADC dose that results in tumor stasis for 21 days after a single administration (a high bar MED. Across models, the MED ranged from <5 to 12 mg/kg. In our cyno safety study, animals were dosed Q3W X 5. Based on these metrics, we calculated a TI of approximately 2-4 for exposure-based, and approximately 4-8 based on mg/m² (BSA). The TI range reflects the use of several tumor models to determine MED. ADCs with the bis-alkylator PBD dimer did not demonstrate a favorable TI (see response to question 3 for details). This information has been added to the manuscript as follows: Therapeutic index (TI) is determined using the MTD (or HNSTD) from nonhuman primate (NHP) safety studies and MED (minimal efficacious dose) from mouse xenograft studies. TI comparison across different molecules is complicated by 1) different definitions of MED (dose that results in any level of tumor growth inhibition vs. dose that results in tumor stasis) and 2) different dosing regimens used in NHP studies. There are several approaches for estimating TI, including the more rigorous determination of TI based on exposure or using body surface area (to account for differences in rodent vs. NHP) vs. simply using MTD and MED based on dose (mg/kg). For TI calculation in our models, we define MED as a single dose that results in tumor stasis for at least 21 days. HNSTD was derived from our safety study with a dosing regimen of Q3W X 5. Given these parameters, the exposure-based (total antibody*

AUC) TI for DHES0815A is 2-4, and the body surface area-based (mg/m²) TI is 4-8. The range reflects using MED from multiple xenograft studies with a range of sensitivity to DHES0815A. In contrast, there is no TI for HER2 ADCs (evaluated at Genentech) with the bis-alkylator PBD payloads, as MED values were ≤ 1 mg/kg and MTD values < 1 mg/kg.

2. In the introduction it is stated that a goal was to provide a different treatment regimen other than T-DM1. I believe they mean an alternative treatment option – to me, regimen implies dosing frequency, e.g. qwk vs. q3wk dosing. Clearly the goal here is different. Response: *Thank you for the suggestion; we completely agree and have replaced 'regimen' with 'option' in the Introduction.*

3. There is an emphasis on reducing the potency, whereas I think the real challenge is increasing the tolerability (although I acknowledge the two are related). Also, it is stated that one reason to aim for a higher tolerated dose is to avoid TMDD, but I believe that even with targets where there isn't TMDD as with HER2, for solid tumors the antibody dose matters. Work by Wittrup, Thurber (e.g. *Journal of Theoretical Biology* 314 (2012) 57–68) highlight this. I would expect that the authors would have no problem with a payload that was picomolar potent as long as it was tolerated at much higher doses in order to allow one to safely dose through any TMDD. Perhaps the authors could better explain their effort as a trade-off on potency in favor of tolerability – i.e. they aren't necessarily increasing the therapeutic index, but rather shifting it into a range that allows for a higher antibody dose (that exceeds the TMDD concern).

Response: *Thank you for highlighting the need for clarification around our objectives. Our goal was several fold: we hypothesized that a reduced potency payload would not only provide a better tolerated ADC but would also allow dosing in a range for optimal PK (i.e., in the linear range, above ~4 mg/kg in cynomolgus monkey, which avoids TMDD). We tested a number of highly potent (pM) PBD payloads and found that, not only were they not tolerated in mice, but efficacious doses and cyno MTDs were ≤ 1 mg/kg, demonstrating no TI for the PBD bis-alkylator. While our efforts demonstrated that we could expand the TI, the authors recognize that this might also result in a shifting of the TI towards higher dose levels in the linear dose range. We added this wording to the section: ≥ 2.5 mg/kg, and rapid clearance at doses below (23). Our prior experience with PBD dimer ADCs demonstrated that MED (minimum efficacious dose in mouse xenograft studies) and MTD (maximum tolerated dose in cynomolgus monkey) were 1 mg/kg or lower. It was thus concluded that PBD ADCs would have a narrow therapeutic index (TI) and would not allow dosing in the linear PK range. In order to*

provide optimal PK, as well as improve the therapeutic index, modifications were made to reduce the potency of the PBD dimer SG2057 (24).

4. *It's stated that only 58% alkylation was observed with the PBD-monoamide". Can the authors comment on why 100% alkylation (or at least closer to 100%) isn't achieved?*

Response: *Thank you for pointing out that a better explanation is warranted for the DNA alkylation assays. We have edited this section to further clarify. The assay uses double stranded oligonucleotides with one G (guanine) per strand, which is the site of alkylation. The bis-alkylator PBD dimer reacts with and alkylates the G in both strands, whereas the monoalkylator PBD-monoamide only reacts with one G. Under these experimental conditions, 100% of spiked double-helix oligo was converted to PBD adducts by incubation with PBD, whereas 58% of spiked oligo was converted to PBD-monoamide adducts after incubation with the monoalkylator PBD-monoamide, as expected. See structure of double-stranded oligo below: :*

```
5'-TATAGAAATCTATA  
  ATATCTTTAGATAT-5'
```

The paragraph was modified as follows: DNA binding assays were performed using double-stranded oligonucleotides with one guanine (G), the site of alkylation, per strand. The bis-alkylator PBD dimer forms adducts with the oligonucleotide by alkylating G in both strands, whereas a monoalkylator reacts with one G. Under these experimental conditions, 99-100% of spiked oligonucleotide was converted to PBD adducts after incubation with the parent PBD dimer and slightly lower alkylation (93%) with the PBD-monoamine. In contrast, 58% of the oligonucleotide was converted to adducts with the PBD-monoamide, in alignment with our goal (Fig. 1b).

5. *In Figure 1a, the ADC units are listed as nM, but the values provided are ug/mL. Ideally the in vitro concentrations for the ADC would be provided in nM to allow for more facile comparison of the impact of antibody delivery*

Response: *We thank the reviewers for catching this oversight. The table in Fig. 1a, as well as Supplementary Fig. 4 have been corrected to show both nM and ng/mL IC50 values for the ADCs.*

6. *Several in vivo efficacy models are utilized, and their HER2 IHC status is indicated. It would be helpful if the antigen expression level for the cell-line derived models could be reported (e.g. is it 500K antigens per cell? 50K? Readers are interested in correlating efficacy with antigen expression)*

Response: *HCC1569 X2 is the only cell line-derived model included in the manuscript for in vivo efficacy studies. Although we have not quantified receptor number on this cell line, it is HER2 3+ by IHC; and FACS histograms overlay with other HER2 3+ lines routinely used (SK-BR-3, KPL-4, which have 1-2 million HER2 per cell). We used PDX models to assess activity in HER2 2+ and HER2 1+ tumors and documented HER2 expression by IHC and ISH. (It would be very challenging to assess receptor number in PDX models which typically do not yield cell lines for in vitro use). Our findings are consistent with data reported in PDX and cell line models for other next generation HER2 ADCs that are active in both HER2+ and HER2-low models, but not in HER2-negative models (Ogitani et al., Clin Cancer Res 2016; Shastri et al., Mol Cancer Ther 2020). As more data emerges, it appears that there is a threshold antigen expression level (HER2 1+) below which HER2 ADCs are not active. Data showing no activity in vitro in HER2-negative (IHC 0) MCF7 and MDA-MB-468 cells are in Fig. 2 and Supplementary Fig. 6, respectively. We did not perform in vivo efficacy studies in a HER2 IHC 0 breast cancer model. To address this, we have included an efficacy study in WSU-DLCL2 B-cell lymphoma model, which does not express HER2, showing no target-independent activity of DHES0815A and dose-dependent activity of CD22 (B cell target) PBD ADCs (Supplementary Fig.14). This study provides information similar to Fig. 1D, which shows lack of target-independent activity of CD22 conjugates in the HER2-expressing Fo5 tumor model.*

7. *In Figure 1D, can the authors comment on why the cell viability for the monoalkylator appears to plateau at 40%? Is this expected given the finding that alkylation only proceeded to ca. 58% (see above)?*

Response: *Thank you for the question and for the opportunity to clarify the cell viability assay results. Cell growth inhibition for the PBD-monoamide conjugate was 40% relative to untreated cells, which represents 60% growth inhibition. This does reflect the finding of 58% DNA alkylation. However, it is unclear if there is a direct correlation between % DNA alkylation and % cell growth inhibition. It can be noted that, even with the PBD dimer conjugate, inhibition did not reach 100%, in contrast to the findings with the free drugs (Fig. 1). It is typical of ADC response in vitro to often not reach 100% growth inhibition, likely due to multiple processing steps of the ADC, asynchronous cell growth, etc. (see Lewis Phillips et al., Cancer Res 2008; Ogitani et al, Clin Cancer Res 2016; Le Joncour et al., Mol Cancer Ther 2019; Barok et al., Cancer Lett 2020 for examples).*

8. *It is indicated that the lack of activity in HER2 negative MCF7 cells indicates target dependence. This would be more convincing if the IC50 of the payload alone in these*

cells was provided, to confirm that these cells are not insensitive / less sensitive to the payload.

Response: We agree that not including potency information for MCF7 cells is an oversight. We repeated the *in vitro* potency assay on MCF7 cells (including several cell lines from the original study as controls for reproducibility) and have included a statement in the manuscript regarding the IC_{50} value for the PBD-monoamide in MCF7 cells. In addition, we tested the free drug PBD-monoamide as well as HER2 and non-targeted PBD-monoamide conjugates in an additional HER2-negative cell line, MDA-MB-468, with similar results as observed with MCF7. The additional data are in Supplementary Fig. 6. Paragraph modified as follows: *In vitro* cell killing assays demonstrated selectivity of DHES0815A for HER2+ BC cells (SK-BR-3), with no activity on HER2-negative MCF7 (Fig. 2a) or MDA-MB-468 cells (Supplementary Fig. 6a). Both MCF7 and MDA-MB-468 are sensitive to unconjugated PBD-monoamide treatment (Supplementary Fig. 6b).

We have also included an *in vivo* study (Supplementary Fig. 14) in the CD22+ WSU-DLCL2 lymphoma model comparing CD22-PBD-monoamide (targeted ADC) with DHES0815A (non-targeted ADC), showing target-dependent tumor inhibition with the CD22 ADC and no target-independent activity with DHES0815A: DHES0815A was compared to a CD22 ADC with the same linker-payload in the CD22-expressing lymphoma model WSU-DLCL2 (HER2-negative). Dose-dependent activity was demonstrated for the CD22 ADC, with no observed efficacy of DHES0815A (Supplementary Fig. 14), indicating no target-independent activity for DHES0815A.

9. The statement “HER2+ breast cancer cells showed differential responses to DHES0815A vs. T-DM1 which correlates with intrinsic sensitivity to the two different payloads...”. Is there any insight into what causes these intrinsic sensitivities? For the monoalkylator a range of close to 1000x is seen between the most sensitive and the least sensitive cell lines – how does this correlate with antigen expression level, Pgp status, or other possible mechanisms of reduced sensitivity / resistance? Is this concerning for the clinical application?

Response: Thank you for the chance to clarify the cell responses to DHES0815A vs. T-DM1. The range of responses to DHES0815A parallels sensitivity to the free drug PBD-monoamide (free drug responses for PBD-monoamide and DM1 are now in Supplemental Figure 7) and is not due to differences in antigen (HER2) expression. None of the cell lines express Pgp or other MDR transporters. We modified this section to include this information: Response to the two ADCs correlated with intrinsic sensitivity to the two different payloads (Supplementary Fig. 7) and was not due

differences in target expression (all cell lines tested were HER2 IHC3+) or expression of drug efflux pumps such as Pgp (cells were negative for Pgp expression). Response to DNA damaging agents is complex and dependent on different types of DNA damage response pathways, which is in contrast to response to agents that bind microtubules. We did not look in detail into which factors contribute to intrinsic sensitivities to the PBD-monoamide. Despite the varied response to DHES0815A in vitro, this ADC was quite active across numerous in vivo models, which lessened concern about clinical application for DHES0815A.

10. The statement “In recent years, it has been appreciated that bystander activity of ADCs is a desirable attribute” – I would propose that this be softened to “may be a desirable attribute”. It is likely highly target and platform dependent.

Response: Yes, we agree and have modified the sentence (In recent years, it has been appreciated that bystander activity of ADCs may be a desirable attribute, depending on target as well as ADC linker-drug). Bystander activity continues to be an area of investigation with different targets and platforms and, for the most part, is not particularly well understood.

11. In Figure 3, the Figure and caption don't indicate the cells used for the experiment. It would be helpful to indicate this clearly in the Figure/legend. This is particularly important given the wide range of sensitivity of different cells as indicated in Figure 2.

Response: We have added text in this paragraph to indicate that these were SK-BR-3 cells, and have added a caption to the figure. (Time-dependent induction of PARP cleavage in SK-BR-3 cells....)

12. For the in vivo efficacy studies, it would be very informative to include IHC images that would provide a perspective on the homogeneity as well as level of HER2 expression

Response: This is a good point as ‘HER2-low’ (IHC 1+ or 2+/ISH-neg) tumors are heterogeneous for HER2 expression. We have added HER2 IHC images for the different in vivo models and methods to the supplemental section (Supplementary Fig. 13).

13. In Figure 4 there are (presumed) errors in the dosing levels – 10 mg/kg is indicated for two lines, whereas no 15 mg/kg line is indicated. Also, Docetaxel is misspelled in 4D.

Response: *Thank you for catching these errors. They have been corrected in Fig. 4 (revised).*

14. *The findings of delayed toxicity are very interesting. However, it is unclear whether what is observed is the result of delayed toxicity or cumulative toxicity. Given the (presumed) irreversible DNA alkylation, do the authors have a view on whether the toxicity is truly delayed (i.e. slow to manifest after a single dose) or rather is the result of cumulative toxicity? Does the tolerability of single, high doses vs. lower multiple doses provide any insight to this? What was the recovery time in the Q3wx2 vs. Q3wx4 studies?*

Response: *The recovery time for the pilot cyno safety studies (Q3wX2, Q3wx4) was 3 weeks. To more fully explore the risk of delayed toxicity signals, the recovery time for the GLP safety study was 7 weeks. This information has been added to the manuscript.*

Due to small numbers of patients, it is difficult to be certain whether the toxicities are “delayed” or a result of cumulative toxicity. Based on the limited data available, it appears that at higher doses, toxicities are observed earlier but they may also be observed later with sustained administration of lower doses at a higher cumulative dose.

Some combination of skin, lung, and ocular toxicities manifested in all patients (n=5) at the 4.0 mg/kg dose resulting in treatment discontinuation for all 5 patients. The median cumulative dose received was 12 mg/kg (range 8.4-19.2). There were two patients who were enrolled into lower dose cohorts (0.6 mg/kg and 1.2 mg/kg) who achieved higher cumulative doses (32.4 mg/kg and 52.8 mg/kg, respectively) due to longer duration of treatment on the basis of clinical benefit. At the time these two patients had a cumulative dose of 12 mg/kg, they did not manifest similar toxicities as seen at the patients in the higher doses. After 30 mg/kg cumulative dose for the first patient (the patient had dose-escalated to 2.4 mg/kg at this time), rash and periorbital edema were noted and after 33.6 mg/kg cumulative dose for the second patient (who remained at 1.2 mg/kg), skin rash and skin hyperpigmentation were noted. The first patient discontinued treatment for clinical progression and the second patient discontinued treatment due to persistence of rash.

The text has been edited to incorporate the additional details about the 2 additional patients who were treated at lower doses with higher cumulative exposure. The text around delayed toxicity was modified to reflect the lack of certainty in the true reason (cumulative exposure vs high dose) for the safety events since there were only 14 patients in the study. See verbatim changes below:

Treatment emergent AEs reported in $\geq 20\%$ of patients are shown in Table 1. Eleven patients (79%) experienced AEs considered related to DHES0815A; the most frequently reported ($n \geq 3$ patients) included pruritus and rash ($n=5$ each, 36%), fatigue ($n=4$, 29%), skin hyperpigmentation, photophobia, and nausea ($n=3$ each, 21%). Following three or more cycles at 4 and 2 or more cycles at 6 mg/kg, multiple safety events involving skin, eyes and lung emerged, often concurrently in a given patient; these events led to treatment discontinuation in all 5 patients treated at these doses. Dermatologic toxicities included pruritus, rash, and skin hyperpigmentation, and were managed with topical and oral antihistamines and corticosteroids. Ocular toxicities included photophobia, conjunctivitis, blepharitis, dry eye, eyelid/periorbital edema, punctate keratitis and eye pain. These toxicities manifested with clinical symptoms and were not identified with routine ophthalmic examinations that were required as part of the study. Management included antibiotics, ophthalmic lubricants, steroids, analgesics, topical anesthetics and other agents. Pneumonitis was reported in 2 patients dosed at or above 4 mg/kg and CT findings of ground glass opacities were reported for both patients. Management included systemic steroids. Due to the small number of patients, it was not clear if any medical intervention for the skin, eye, and lung toxicities improved the severity or duration of symptoms; all 5 patients had unresolved sequelae from some of these toxicities at the time of discontinuation from study follow-up. Supplementary Fig. 16 depicts rash and pulmonary findings and outlines the time course of AEs in a patient who was treated at 4.0 mg/kg for 3 cycles. Two patients assigned to lower doses (0.6 mg/kg and 1.2 mg/kg doses) received a higher cumulative dose compared to any patient enrolled at doses of ≥ 4.0 mg/kg over the course of 1.4 years and 2.6 years, respectively. At the time treatment was discontinued for these patients, one patient had developed rash and periorbital edema and the other had developed rash and skin hyperpigmentation. While patient numbers are limited, the higher cumulative doses tolerated by these two patients compared to the patients at doses of ≥ 4.0 mg/kg suggests that safety events are not only related to cumulative exposure but likely also related to the maximum dose administered in a single infusion.

Despite promising anti-tumor activity, the severity, persistence, and non-resolvable nature of the toxicities compelled the decision to discontinue this phase 1 trial. Due to the limited responses at doses < 4.0 mg/kg (1 patient at 1.2 mg/kg), it was deemed unlikely to identify a dose with sufficient efficacy and safety to warrant additional development in the clinic.

15. The authors are to be applauded for including NHP tolerability data, as this is rarely shared in publications. Given that only one animal showed focal alveolar degeneration, I would suggest that the statement that these findings were reversible as they weren't

observed in recovery animals is perhaps an overstatement; it may be that this is a less frequent finding and is not reversible. Any comments on whether certain toxicities are considered target related or not would also be welcomed, and it would be ideal if a therapeutic index (range) based on AUC for NHP tolerability vs. AUC mouse efficacy would be given.

Response: *We appreciate these comments on the NHP study. The sentence(s) regarding reversibility have been modified to state that the findings were reversible, with the caveat that the findings occurred in only one animal and may be infrequent findings in NHP. We have added that observed toxicities were considered not target related as similar safety signals were observed with targets other than HER2 and with non-targeted ADCs (findings from pilot safety studies). Text edited as follows: Although these findings were reversible as they were not observed in the recovery animals, it is possible that the focal alveolar degeneration is an infrequent finding that might be more prevalent, and potentially not reversible, with larger numbers of animals. The observed safety signals were considered target-independent, as similar safety findings were observed for other ADCs, including non-targeted ADCs, with PBD-type payloads. Please see response to question #1 regarding TI.*

16. In the summary of clinical responses for the 14 patients, data is only described for 13 patients (one CR, 10 SD and 2 PD). What about the 14th patient? Also, in the waterfall plot (Fig 6A) there are more patients that achieved >30% reduction in tumor volume, however they don't qualify as PRs by RECIST. Is this because the tumor reduction was not confirmed on the following scan, or for other reasons? A sentence to explain this would be helpful. Also, the HER2 IHC status is indicated as 2+ or 3+, but there is no indication of the FISH status. It would be helpful to indicate whether or not these patients are considered HER2-positive as opposed to just indicating the IHC score, as IHC-2+ is ambiguous with regard to HER2-positivity by current diagnostic standards.

Response: *There was an error in the initial submission. There was 1 patient with complete response, 2 patients with partial response, 1 patient considered not-evaluable for response, 8 patients with SD, and 2 patients with PD. This was corrected as follows: Out of 14 evaluable patients, one patient (7%) dosed at 1.2 mg/kg experienced a confirmed complete response after 6 cycles according to RECIST v1.1 (34). Two patients achieved a partial response (14%), eight patients (57%) showed a confirmed best response of stable disease, two had progressive disease (14%) and one patient was not considered evaluable by the investigator (7%).*

The patient who achieved -38% reduction had a new lesion, leading to PD. This was clarified in the figure.

All patients enrolled were HER2-positive by the 2013 ASCO/CAP guidelines, which were contemporary with the trial enrollment timelines. Thus all patients who were IHC-2+ were HER2+ by FISH. This was clarified as follows: All patients were required to have HER2-positive disease as defined by ASCO/CAP 2013 guidelines at the time of enrollment (33).

17. I do believe that more detail is required in the methods; for example, at a minimum the specific cell lines used for in vitro experiments need to be disclosed. As noted above, for some experiments (such as the caspase activation) it is not clear which cell line was used.

Response: *We have added this information to the Methods section of the manuscript as follows: For free drug assays, cells tested were SW900 and NCI-H1781 NSCLC lines, BJAB lymphoma, MES-SA and MES-SA/Dx5 (expresses Pgp) uterine sarcoma, and KPL-4, T-47D, HCC1937 and HCC1560 X2 breast cancer lines. Cells were treated for 4 days with different PBD derivatives. Experiments for ADC potency were performed on breast cancer cell lines SK-BR-3, KPL-4, BT-474, HCC1569, HCC1954 and MCF7 treated for 5 days.*

Caspase activation assays were performed on SK-BR-3 and KPL-4 cells. ADCC assays using SK-BR-3 and KPL-4 cells were performed with purified human PBMCs and tumor cells as previously described (7).

Reviewer #2:

1. As an ADC that targets HER2-expression is an already well-established therapeutic strategy in widespread use, and this study of DHES0815A does not clearly demonstrate any evidence of clear superiority of this agent to others available (efficacy & toxicity), we assume this study will not be of great interest to the NC Readership.

Response: *At the time we started preclinical development of DHES0815A, there was only one approved HER2-directed ADC (T-DM1). DHES0815A was clearly differentiated from T-DM1 in terms of structure (antibody-linker-drug) as well as better efficacy vs. T-DM1 across preclinical models. Safety signals were different than T-DM1, due to the different MOA of the payload, with a TI similar to or better than T-DM1. Design of the ADC was intended to allow combination with other HER2-directed*

therapies as well, as the majority of HER2 ADCs use trastuzumab as the antibody backbone. During development of DHES0815A, preclinical data and early clinical information were released regarding T-DXd (trastuzumab deruxtecan). Because of our strong scientific rationale for designing DHES0815A, we had every expectation that this ADC would be competitive with T-DXd. Moreover, the PBD dimer class of cytotoxic payloads has been investigated for years, with only one approved ADC (loncastuximab tesirine) in a hematologic malignancy (DLBCL). Lack of success, especially in solid tumor indications, has largely been attributed to the very high potency of this payload class, resulting in inability to dose to efficacy without tolerability issues. Our hypothesis for using a reduced potency PBD dimer arose to address these specific issues. Our publication of details regarding rational drug design, preclinical safety and clinical safety/efficacy will likely be of great interest to NC and other audiences as well, and will serve to inform the ADC community, most notably because this type of detailed information is rarely disclosed when an ADC is terminated(for preclinical reasons or in phase 1).

2. For HER2 ADCs, it is now known that lower toxic Payload and higher DAR (drug-to-antibody ratio) may have more chances to be successful. Highly toxic Payload is often difficult to achieve the expected efficacy, and will also produce more risks of delayed toxicity. Although this article contains a lot of information, such as design of DHES0815A, mechanisms of action, efficiency in HER2+and HER2-low xenograft models, safety studies in non-human principles, and phase 1 safety and efficiency data, data volume is large but the results are not ideal.

Response: Development and approval of trastuzumab deruxtecan is the first, and only, example of a high DAR HER2 ADC with a lower potency payload. Published reports on the design of T-DXd did not describe DXd as a low potency payload (it was described only as more potent than the parent compound SN-38); in fact, numerous conference presentations on T-DXd describe the payload as 'highly potent' (which is not entirely accurate). Thus it is fair to conclude that selection of a lower potency payload was not the rationale for the design of T-DXd. We initiated development of DHES0815A with this specifically in mind – use of a lower potency payload to improve the TI by achieving higher doses for optimal efficacy and PK. Despite a strong and compelling preclinical package, phase 1 results were disappointing and, we agree, not ideal. However, that is our intent for publishing our findings. It is rare that the challenging drug development stories are told. That our end result was not ideal is a primary reason for publishing this manuscript in that these types of learning made public will benefit others developing ADCs.

3. The author did not fully discuss the reasons for the failure of this research and the implications for the future. As authors mentioned, late-onset adverse events, especially those related to the accumulation of dosing, led to close DHES0815A trial, but DS8201 with interstitial lung disease is a success story. A failure of a novel agent may be dependent on other factors, including the modification of anti-HER2 antibody and conjugation method, linker technology (disulfide linker more likely to lead to fall-off of payload in non-targeted tissue), besides the payload cause. Please discuss more on reasons for DHES0815A failure, details of the late-onset AE reported in this manuscript, for example, dyspnea: any CT finding, and presence or absence of ILD? Headache: what's the cause? Any imaging finding? Photophobia: any ocular test results? More important, information on medical interventions and clinical outcomes for those AEs are much needed.

Response:

For DHES0815A, the only modification to the antibody was substitution of cysteine for lysine on light chain site 149 to enable site-specific conjugation (THIOMAB technology) on the 2 light chains. Binding of THIOMAB antibodies to target antigens has been demonstrated to be equal to target binding of parent wildtype antibodies. Disulfide linker technology has vastly improved since the first linkers described more than 2 decades ago. The hindered disulfide linker on LC K149C (in DHES0815A) is highly stable for ADCs with PBD payloads (Pillow et al., ref. #20). For these reasons, we conclude that the toxicities observed are due to the nature of the payload and not to antibody modifications, conjugation method or linker instability.

Clinical development of DHES0815A was discontinued due to the nature of the toxicities observed. For the 5 patients who were treated at doses of 4.0 mg/kg and higher, 4 patients experienced skin and eye toxicity events related to study treatment which were persistent for several weeks even after treatment discontinuation. Ophthalmic examinations were performed for all patients; punctate keratitis, conjunctivitis and blepharitis were noted in one patient, conjunctivitis alone was noted for one patient, and periorbital edema alone was noted in 2 patients. A variety of interventions were tried including antibiotics, ophthalmic lubricants/wetting agents, steroids, analgesics and topical anesthetics, but it was not clear if any of these interventions were helpful in improving symptoms or recovery time. In some cases, toxicities remained unresolved at the time of discontinuation from study follow-up; in other cases, toxicities persisted for up to 5 months or longer from onset. Pneumonitis was seen in 2 patients treated at the higher doses; CT findings of ground glass opacities were reported. Systemic steroids were administered. Neither case was considered resolved at the time of discontinuation from study follow up. Imaging for headache was not performed but in the 3 patients who developed headache at the higher doses, the headache was attributed to the

concurrent ocular toxicities. Manuscript was edited as follows: Pneumonitis was reported in 2 patients dosed at or above 4 mg/kg and CT findings of ground glass opacities were reported for both patients. Management included systemic steroids. Due to the small number of patients, it was not clear if any medical intervention for the skin, eye, and lung toxicities improved the severity or duration of symptoms; all 5 patients had unresolved sequelae from some of these toxicities at the time of discontinuation from study follow-up. Supplementary Fig. 16 depicts rash and pulmonary findings and outlines the time course of AEs in a patient who was treated at 4.0 mg/kg for 3 cycles. Two patients assigned to lower doses (0.6 mg/kg and 1.2 mg/kg doses) received a higher cumulative dose compared to any patient enrolled at doses of ≥ 4.0 mg/kg over the course of 1.4 years and 2.6 years, respectively. At the time treatment was discontinued for these patients, one patient had developed rash and periorbital edema and the other had developed rash and skin hyperpigmentation. While patient numbers are limited, the higher cumulative doses tolerated by these two patients compared to the patients at doses of ≥ 4.0 mg/kg suggests that safety events are not only related to cumulative exposure but likely also related to the maximum dose administered in a single infusion.

Objective response at doses lower than 4 mg/kg was limited with CR in one patient at 1.2 mg/kg. Due to limited responses at lower doses, it was determined there would not be a dose with sufficient efficacy and safety to warrant additional clinical development. In addition, the severity and persistence of the observed toxicities led to the decision not to explore additional doses under 4 mg/kg or alternative dosing schedules (e.g., decreased dosing frequency).

The preliminary efficacy observed suggests that the antibody used was able to direct payload to HER2-bearing tumor cells. There were also very low levels of free payload detected in serum. These findings led to the conclusion that the payload rather than antibody and linker are likely responsible for the unfavorable risk/benefit observed in the clinical study.

4. We agree that the recent phase I trial design is impossible to define the late-onset fatal AE in the first 28 days after drug administration, but it can be further characterized in the following phase II and III trial

Response: We would like to clarify that there was no late-onset fatal AE in this study. AEs were documented after patients cleared the DLT window of 1 cycle (21 days). The phase 1 was discontinued due to AEs; there will be no phase 2 or 3 studies.

Reviewer #3:

1. A mechanistic description/hypothesis explaining why such toxicity occurred later on study and why it was missed, or at least underestimated, in previous phases of development would strengthen the paper, making it more scientifically attractive.

Response: *Thank you for the chance to further elucidate on our findings and hypotheses for the delayed toxicities. The nature of some of the toxicities were similar to findings from the NHP cyno study (skin hyperpigmentation, corneal pigmentation). After a recovery period of 7 weeks, these toxicity events, along with modest changes in hematology parameters and lung findings in 1 animal, were reversible. We believe that the general mechanism underlying the safety signals is accumulation of DNA damage after prolonged tissue exposure to the monoalkylator payload. Although the PBD-monoamide does not cross-link DNA, alkylation does result in DNA damage, which can accumulate over time after multiple doses of an agent (such as an ADC) that has a long half-life. The health of patients with metastatic breast cancer who have received multiple prior therapies is understandably compromised; thus, the patients in our study were highly sensitive to the toxic effects of the ADC compared to what was observed in the NHP preclinical safety study. We have modified the discussion to reflect our thinking as follows: Although most DHES0815A-related AEs were categorized as grade 1-2, the severity, persistence, and non-resolvable nature of the toxicities observed at doses of 4.0 mg/kg and higher compelled us to reduce the dose of patients still on treatment to 2.4 mg/kg, discontinue enrollment, and ultimately close the trial. Our hypothesis regarding the observed toxicities is that, although PBD-monoamide does not cross-link DNA, monoalkylation can result in DNA damage, most notably after repeat dosing. At the higher doses of DHES0815A, DNA damage likely accumulates in certain tissues, resulting in toxicity signals. However, it was surprising that the nature of these toxicities in patients was marked, compared to cynomolgus monkeys, given that we allowed for an extended recovery period in the cynomolgus monkey safety study.*

Reviewers' Comments:

Reviewer #1:

Remarks to the Author:

I appreciate the responses to the questions / suggestions I posed in my initial review, and I am satisfied with the answers and the corrections to the manuscript. I congratulate the authors on the work and the manuscript, and I do believe it will be of high interest to the ADC scientific community and beyond. I am in favor of publishing the manuscript in its revised form.